# Plasticity of muscle synergies through fractionation and merging during development and training of human runners

Vincent C. K. Cheung 1,2,7✉, Ben M. F. Cheung 1,7, Janet H. Zhang 3,4, Zoe Y. S. Chan3,5, Sophia C. W. Ha1, Chao-Ying Chen 3 & Roy T. H. Cheung3,6✉

Complex motor commands for human locomotion are generated through the combination of motor modules representable as muscle synergies. Recent data have argued that muscle synergies are inborn or determined early in life, but development of the neuro-musculoskeletal system and acquisition of new skills may demand fine-tuning or reshaping of the early synergies. We seek to understand how locomotor synergies change during development and training by studying the synergies for running in preschoolers and diverse adults from sedentary subjects to elite marathoners, totaling 63 subjects assessed over 100 sessions. During development, synergies are fractionated into units with fewer muscles. As adults train to run, specific synergies coalesce to become merged synergies. Presences of specific synergy-merging patterns correlate with enhanced or reduced running efficiency. Fractionation and merging of muscle synergies may be a mechanism for modifying early motor modules (Nature) to accommodate the changing limb biomechanics and influences from sensorimotor training (Nurture).

[1] School of Biomedical Sciences, and The Gerald Choa Neuroscience Centre, The Chinese University of Hong Kong, Hong Kong, China. [2] Joint Laboratory of Bioresources and Molecular Research of Common Diseases, The Chinese University of Hong Kong and Kunming Institute of Zoology of The Chinese Academy of Sciences, Hong Kong, China. [3] Gait & Motion Analysis Laboratory, Department of Rehabilitation Sciences, The Hong Kong Polytechnic University, Hong Kong, China. [4] Department of Integrative Physiology, University of Colorado, Boulder, CO, USA. [5] Faculty of Kinesiology, University of Calgary, Calgary, AB, Canada. [6] School of Health Sciences, Western Sydney University, Sydney, NSW, Australia. [7] These authors contributed equally: Vincent C. K. Cheung, Ben M. F. Cheung. ✉email: vckc@cuhk.edu.hk; Roy.Cheung@westernsydney.edu.au

To generate a movement, the central nervous system (CNS) must compute coordinated commands for thousands of motor units within hundreds of skeletal muscles. For the movement to occur as intended, not only does the CNS need to specify a tremendous number of output variables, it must also take into account, during this specification, various biomechanical demands or constraints imposed on movement execution. One source of constraints comes from the current properties of the neuro-musculoskeletal system, including muscle strength, muscle stiffness, gains of neuromotor reflexes etc.[1]. Thus, the muscle commands required for a preschooler to accomplish a movement, for instance, can be very different from those required for an adult because of differences in the biomechanical design of their limbs. Another obvious source of biomechanical demands comes from constraints imposed by the motor task itself. The learning of a new, difficult skill with novel kinematic, kinetic, or energetic requirements, for instance, would likely necessitate the acquisition of new muscle coordination patterns. The CNS must be equipped with mechanisms that can efficiently adjust motor commands, over multiple time scales, to accommodate both the changing neuro-musculoskeletal properties during development[2] and the changing task demands during learning, while ensuring that the high-dimensional commands are spatiotemporally coordinated and robustly generated.

The above considerations suggest that the motor-output generating neuronal networks must be versatile enough to satisfy the everchanging biomechanical needs of motor control. Such flexibility demanded of the CNS is seemingly at odds with the fact that any motor command produced must be permissible by the structures of the underlying neuronal networks – or the neural constraints on movement generation[3–5]. It has been argued that these constraints assume a modular architecture that facilitates the coordination and execution of diverse motor behaviors[6]. Recent studies have shown that for at least some of these modular constraints, called motor primitives or motor modules, their structures appear to be either inborn[7] or determined very early in life[8]. Indeed, many of them remain rather invariant over most of the life span[5,8] or even across species[7]. But other data have also indicated that the individual's motor development and the acquisition of difficult skills may only be possible after new modules are acquired[7,9,10] or after the preexisting modules are modified[8,11–13]. By what principles the early motor modules from Nature are updated by the CNS to fulfill the needs arising from development and learning – i.e., influences from Nurture – is a fundamental question in neuroscience.

We seek to shed light on the malleability of the neural constraints on movement during motor development and learning by studying the motor patterns of running in human subjects at different developmental and training stages. Children and adults alike can run without prior training. Yet, the biomechanics of running for these two age groups are very different[14]. At the same time, for adults, performance running that maximizes running economy is a skill that demands motor training[15]. Thus, running, a fundamental form of human locomotion, is an ideal in-born motor behavior[16] to study for addressing our question of how neural constraints are adjusted during development and learning.

To represent and characterize the motor modules, we note that muscles are activated by spinal motoneurons, and activities of the motoneurons of multiple muscles are coordinated by networks of spinal premotor interneurons[17–19] and motor cortical neurons[20]. The structures of these higher-order networks facilitate motor coordination by activating specific groups of muscles together as modules that implement task-relevant biomechanical functions[21], thereby acting as neural constraints on movement by reducing the search space of motor commands[22]. Accumulated evidence has argued that a motor module can be represented as a muscle synergy – a time-invariant, multi-muscle activation balance profile that is scaled by a time-varying coefficient, and that linear summation of the activations of a manageable number of muscle synergies can explain the observed variability of multi-muscle patterns (Fig. 1). Muscle synergies and their temporal activations can be identified by applying the non-negative matrix factorization (NMF) algorithm to electromyographic data (EMG), which record activities of the muscles resulting from motoneuronal activations. Importantly, it has been shown recently that putative muscle synergies extracted from multi-muscle EMGs using NMF correspond well to how spinal interneuronal networks co-activate the motoneuronal pools of multiple muscles[19]. Thus, to address our question on neural constraints, we first recorded, from multiple subject groups at different ages and with varying running experience, surface EMGs (15 right-sided lower-limb muscles) during running at their self-selected preferred speeds (Fig. 1). We then examined how the NMF-derived muscle synergies for running[23–25] differed across groups with both cross-sectional and longitudinal comparisons.

Our comparisons reveal that muscle synergies for running exhibit considerable developmental and training-related plasticity. During child-to-adult development, some early muscle synergies fractionate into units with fewer muscles. During adult running training, some pre-training synergies coalesce into merged synergies. Across adults, presences of specific patterns of synergy merging correlate with enhanced or reduced energetic efficiency of running. Our results argue that muscle synergies can undergo long-term reorganization to fulfill the changing biomechanical and functional needs of locomotion.

## Results

**Strategy for revealing muscle-synergy plasticity**. To study the changes in the muscle synergies through the developmental and training stages, we adopted a combined cross-sectional-longitudinal design by recording EMGs from five groups – preschoolers (Presch; age 3–6), sedentary adults without ongoing and previous training (Sedent), novice adult runners in training (Novice), as well as experienced (Exp) and elite runners (Elite) – and by recording multiple time points at the same preferred running speed for each subject from the sedentary (at 0 and 2 months; Sedent0 and Sedent2) and novice (0, 3 and 6 months into training; Novice0, Novice3 and Novice6) adults (Fig. 1a). A comparison between Presch and Sedent would highlight changes attributable to child-to-adult development without too much confounding adult running experience because the latter had no prior training. A comparison across the adult groups, from the Sedent and Novice time points to Exp and Elite, would highlight gradual changes associated with the acquisition of expertise over months to years. Indeed, all elites could run a sub-3-h marathon.

**Developmental fractionation of early muscle synergies**. We began with a comparison of the Presch and Sedent0 muscle-synergy sets by characterizing their dimensionalities (i.e., the number of muscle synergies composing the EMGs) and synergy-vector similarity. For an EMG-reconstruction $R^2$ of ~80%, the Presch data demanded a smaller number of synergies than Sedent0 (~6 vs. ~7; Fig. 2a). The muscle synergies from all subjects of each group were then clustered by $k$-means, and we identified nine clusters in Presch, and 12 in Sedent0 (see Supplementary Table 1 for the muscles in the Sedent0 clusters). For both Presch and Sedent0, we then identified the subject-invariant clusters by noting those with synergies contributed by >1/3 of the group, and found seven subject-invariant clusters in Presch, and 11 in Sedent0 (Fig. 3a). When the subject-invariant clusters of these two groups were compared against each other, 6 synergy

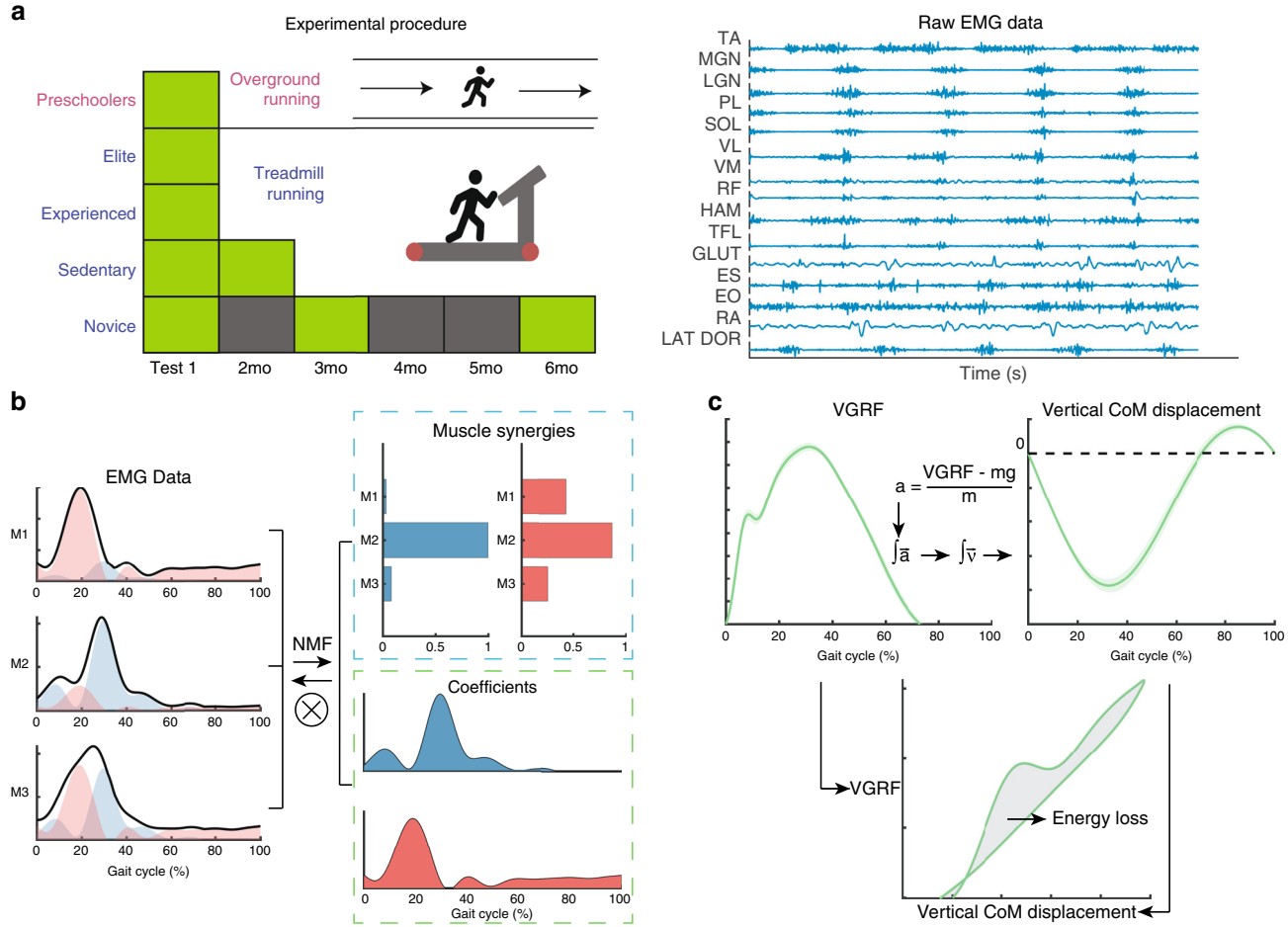

**Fig. 1 A schematic of experimental design. a** Preschoolers and four groups of adults (sedentary, novice, experienced, and elite runners) were studied. EMGs (15 muscles) were collected during over-ground (preschoolers) or treadmill (all adults) running, over 1–3 sessions. **b** Pre-processed EMGs of every session from each subject were decomposed into time-invariant muscle synergies activated by time-varying coefficients using the non-negative matrix factorization algorithm (NMF). The NMF captures the multivariate structure of variability embedded in EMGs. **c** Vertical ground reaction force (VGRF) of each running cycle was recorded for deriving, together with body mass and weight (m*g), the acceleration (a), the double integration of which yielded the vertical center-of-mass (CoM) displacement (see Methods for the assumption used for determining the integration constant). A plot of the VGRF against the CoM displacement produced a hysteresis loop, the area within which provided an estimate of the mechanical energy loss during each cycle. Source data for **b** and **c** are available as a Source Data file.

pairs from the two groups could be matched with moderate to excellent similarity (scalar product, or SP ≥ 0.87), but 5 Sedent0 clusters could not be well matched to any Presch cluster (SP < 0.8). Notably, for the moderately well-matched pairs (Fig. 3a, pairs 4–6), the Sedent0 synergies had fewer active muscle components than their Presch counterparts. Also, all unmatched Sedent0 synergies tended to have significant activation components in only 1–2 muscles. Quantification of muscle-synergy sparseness confirmed that the Sedent0 synergies were significantly sparser than those of Presch (Fig. 2b).

The above observations suggest that, while ~6 muscle synergies were partially to fully preserved from Presch to Sedent0, Sedent0 also had more unique varieties of synergy that were more fragmented in their muscular compositions. To elucidate the origin of these fragments, we posited that some Sedent0 synergies may emerge from the fractionation of specific Presch synergies, in the sense that a linear combination of the Sedent0 synergy fragments can reproduce the original split Presch muscle synergy[26]. With this model, we found that three of the Presch muscle-synergy cluster centroids – in fact, the same ones that were moderately well-matched to Sedent0 clusters (Fig. 3a, P-4, -5, and -6), and all primarily involving extensors – could be well explained

(SP ≥ 0.93) as synergies that were fractionated to give rise to 6 Sedent0 clusters (Fig. 3b). Specifically, all three of these Presch synergies produced a tibialis anterior (TA)-only fractionation (S0-7), and two produced one with tensor fasciae latae (TFL) and gluteus maximus (GLUT) (S0-8). This pattern of fractionation was further validated by identifying fractionation instances in the muscle-synergy sets of every Presch-Sedent0 subject pair (Fig. 3c). Thus, child-to-adult development of motor patterns for running is underpinned in part by fractionation of the early muscle synergies observed in preschoolers. The major patterns of developmental synergy fractionation are summarized in Supplementary Table 2.

**Merging of pre-training muscle synergies during training.** If child-to-adult development of running is associated with synergy fractionation, we wondered whether training on running for adults leads to further fractionation or not. Across the adult groups from Sedent0 to Elite, we noted a decrease in the number of muscle synergies (from ~7 to ~6; Fig. 2a). Accompanying this decrease in EMG dimensionality was a decrease in the sparseness of the muscle-synergy vectors from Sedent0 to Elite (Fig. 2b), implying that, the more experienced the runner, the more

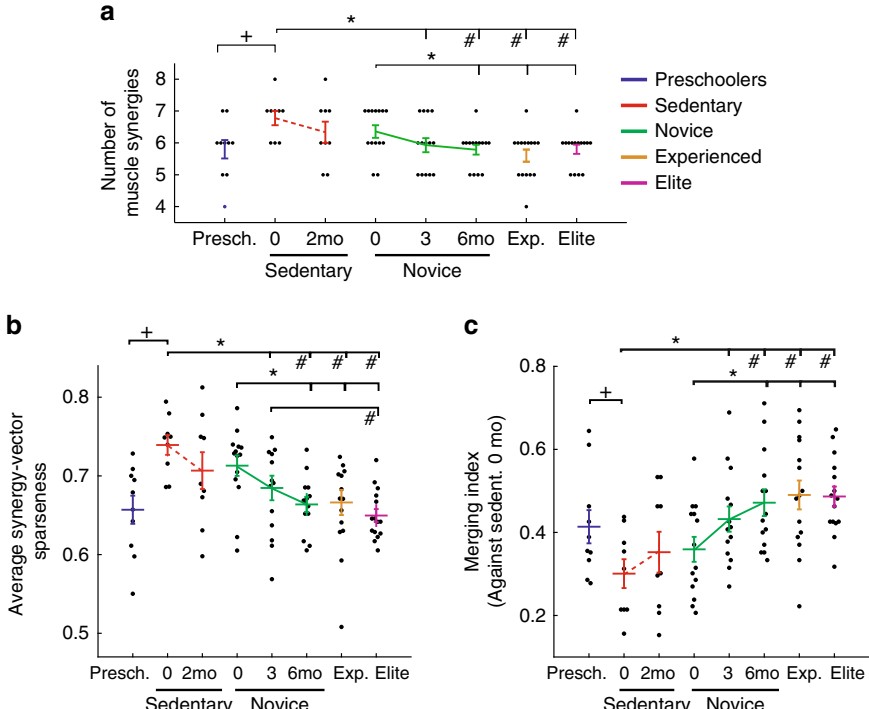

**Fig. 2 General characterizations of the muscle synergies across subject groups. a** The number of muscle synergies (dimensionality) increased from Presch (~6) to Sedent0 (~7) (+, $p = 0.010$, 2-tailed Mann–Whitney; mean ± SE), and decreased from Sedent0 to Novice/Exp/Elite (~6) ($p = 0.0058$, Kruskal–Wallis; #significant after multiple comparison of all adult groups at $\alpha = 0.059$; *$p < 0.05$, 2-tailed Mann–Whitney). A statistically significant change in this number was also found across the Novice (solid line, $p = 0.046$, Friedman) but not the Sedent time points (dotted line). **b** The average sparseness of the muscle-synergy vectors (mean ± SE) was quantified for every condition. From Presch to Sedent0, an increase in sparseness was observed (+$p = 0.0018$, 2-tailed t-test); but from Sedent0 to Novice/Exp/Elite, a decrease ($p = 0.0003$, Kruskal–Wallis; #significant after multiple comparison at $\alpha = 0.05$; *$p < 0.05$, 2-tailed t-test). A significant change was also found across the Novice (solid line, $p = 0.017$, Friedman) but not the Sedent time points. **c** Results in **a** and **b** suggest that the Sedent subjects may possess the most fundamental set of synergy vectors whose merging could explain the synergies of other conditions. We devised a Merging Index (MI) (mean ± SE) that quantifies, for each subject, the percentage of synergies explainable by merging Sedent0 synergies. The MI decreased from Presch to Sedent0 (+$p = 0.0499$, 2-tailed t-test). There was a notable increase in MI from Sedent to Novice/ Exp/Elite ($p = 0.0003$, ANOVA; #significant after multiple comparison; *$p < 0.05$, 2-tailed t-test). A significant change was found across the Novice (solid line, $p = 0.037$, rm-ANOVA) but not the Sedent time points. For all panels, sample sizes of the five groups were, $n = 10$ (Presch), 9 (Sedent), 14 (Novice), 15 (Exp), and 15 (Elite). Source data for all panels are available as a Source Data file.

number of active muscles there were in each synergy on average. These observations suggest that, instead of fractionation, training on running leads to merging of specific muscle synergies. Here, merging is understood in the sense that a synergy in the more experienced groups that results from merging can be constructed by linearly combining the to-be-merged synergies in the least experienced Sedent0 group. We devised a procedure for detecting instances of synergy merging between any subject pair, analogous to the one used for detecting fractionations. To quantify the extent of synergy merging across groups, we formulated a "Merging Index" (MI), which measures the percentage of synergies of a subject explainable by merging multiple synergies from subjects in Sedent0. From Sedent0/2 to Novice0/3/6 to Exp and Elite, the MI increased gradually (Fig. 2c). Thus, changes in muscle pattern as adults train on running are underlain by the merging of the pre-training muscle synergies (see Supplementary Note 1, Supplementary Fig. 1, Supplementary Table 3).

Consistent with the patterns of synergy fractionation and merging shown above, among the subject groups, Sedent0 had synergies that were most generalizable in terms of being able to explain the variances of the EMGs of other groups (Supplementary Note 2, Supplementary Fig. 2),.

**Synergy merging patterns related to running efficiency.** In the adult groups, in addition to muscle synergies, we estimated each

subject's energetic efficiency of running (Fig. 1C; Supplementary Note 3, Supplementary Fig. 3A). Not surprisingly, the more experienced the subject, the more energetically efficiently the subject tended to run (true except for Exp; Supplementary Fig. 3B). It is reasonable to expect that across adults, the merging of specific muscle synergies may enable the subject to run more efficiently. To discover these potentially energetically relevant merging patterns, we identified all instances of synergy merging in all adults using the Sedent0 synergies as the basis vectors, and found 189 unique merging combinations. After filtering out those with ≤30% prevalence in all groups, there were 27 potentially functionally important combinations, for each of which the adults possessing the combination in their synergy sets (the have's) were compared against those not having it (the have-not's) in their running efficiency values.

In our comparisons, we found five merging combinations for which the have's and have-not's showed a statistically significant difference in running efficiency ($p = 0.011–0.049$, 2-tailed Mann–Whitney). Of these five, there were three (Fig. 4a) such that the have's had a higher average energetic efficiency of running than the have-not's (Fig. 4b). Denoting the Sedent0 cluster number (Fig. 3a) by the prefix S0, these combinations were, S0- 7 + 11 (i.e., merging of clusters S0-7 and S0-11), 5 + 6 + 8, and 5 + 6 + 12. Remarkably, for all three combinations, their frequencies of occurrence in subject groups increased from the

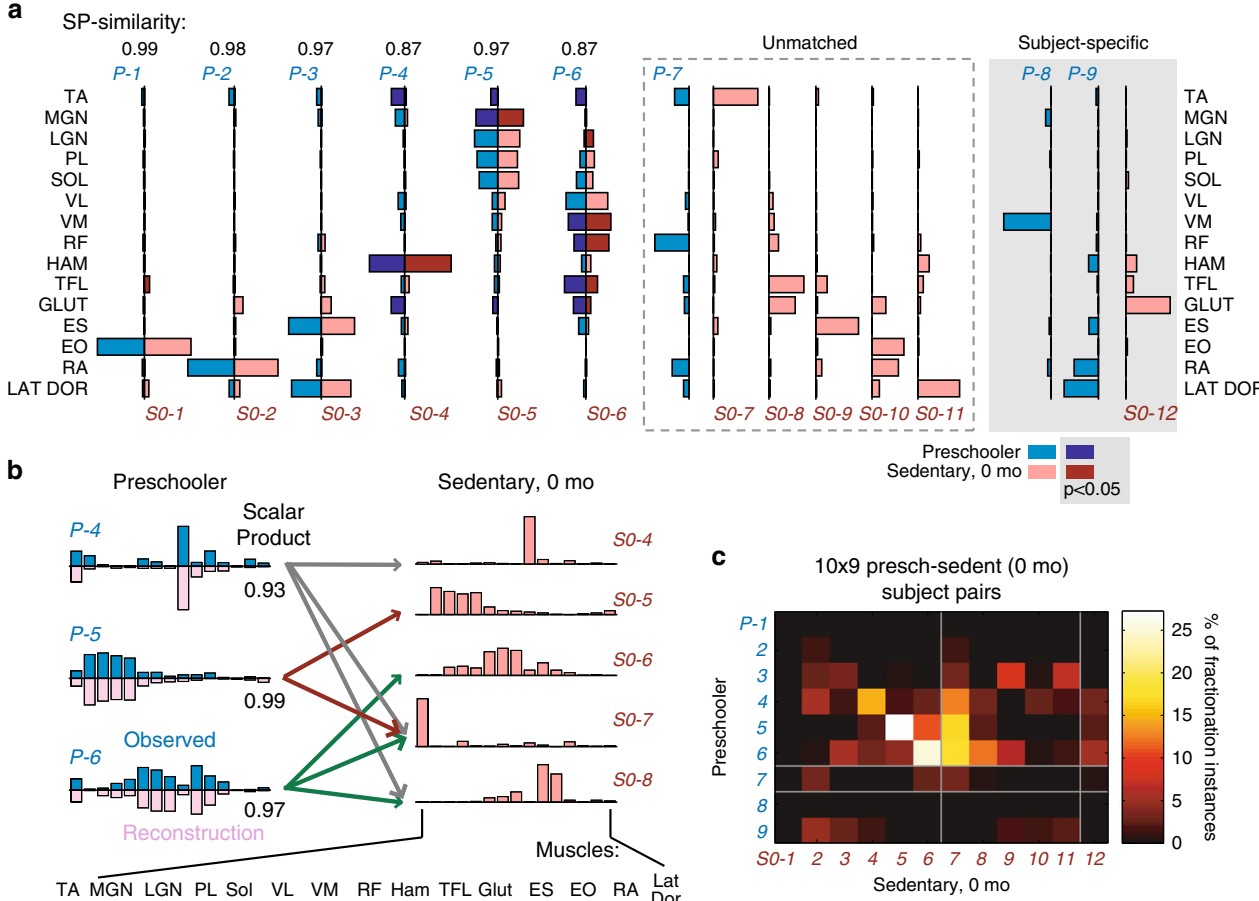

**Fig. 3 Fractionation of muscle synergies during motor development. a** Changes of muscle synergies for running associated with motor development were studied by comparing the Presch and Sedent0 synergies. Shown are the *k*-means muscle-synergy cluster centroids (blue, Presch, *P*-1 to 9; pink, Sedent0, *S0*-1 to 12), matched by maximizing scalar product (SP). After matching, the component values of every muscle of each cluster were compared between the two groups (dark blue and red, *p* < 0.05; 2-tailed t-test). The subject-specific clusters – those with synergies contributed by <1/3 of the group – were excluded from matching. We note that six cluster pairs (*P*- and *S0*-1 to 6) were moderate-to-well matched (SP between centroids = 0.87–0.99). *P* values for cluster 4: TA, 0.043; HAM, 0.0046; GLUT, 0.016. *P* values for cluster 5: TA, 0.016; MGN, 0.015. *P* values for cluster 6: TA, 0.0047; LGN, 0.0001; VM, 0.0056; RF, 0.0072; TFL, 0.012; GLUT, 0.035. Numbers of synergies within the clusters were, from *P*-1 to 9, n = 10, 4, 7, 8, 10, 10, 5, 1, 3, and from *S0*-1 to 12, n = 5, 5, 4, 5, 9, 8, 9, 3, 4, 4, 3, 2. **b** Presch cluster centroids *P*-4 to 6 could be explained by linearly combining multiple Sedent0 centroids (SP between original and reconstructed vectors = 0.93–0.99). Thus, each Presch synergy here was fractionated to become multiple synergies in Sedent0 (e.g., *P*-4 was split to *S0*-4, 7 and 8). Arrows denote the pattern of fractionation (gray, for *P*-4; red, *P*-5; green, *P*-6). **c** Synergy fractionation patterns were further characterized by comparing the synergy sets of every of the 90 Presch/Sedent0 subject pairs. Shown is a heat map depicting the percentage of all detected fractionation instances that involved any Presch (*P*-) cluster producing a fractionation that belonged to any Sedent0 (*S0*-) cluster. Most instances involved *P*-4 to 6 fractionated into *S0*-4 to 8. Gray horizontal and vertical lines separate the matched, unmatched, and subject-specific clusters for Presch and Sedent0, respectively, as indicated in **a**. Source data for all panels are available as a Source Data file.

least to most experienced adult groups (*r* = 0.83–0.90; Fig. 4c), suggesting that their emergence may be a consequence of running training (see also Supplementary Note 4, Supplementary Fig. 4).

The other two energetically relevant merging combinations – *S0*- 3 + 12 and 4 + 5 + 7 (Fig. 5a) – were associated with a lower efficiency in the have's than the haven-not's (Fig. 5b) among either all adults (*S0*-3 + 12) or the Exp and Elite subjects (*S0*-4 + 5 + 7). Combination *S0*-4 + 5 + 7 was observed in only a small percentage of subjects in all groups except Exp, but combination *S0*-3 + 12 was more prevalent in Sedent0/2 than the more experienced Novice0/3/6, Exp, and Elite (Fig. 5c). This result suggests that running training may necessitate learned suppression of this merging pattern whose presence correlated with reduced running efficiency.

**Merging combinations required for higher running efficiency.** Above, we have identified synergy merging combinations associated

with either increased (*S0*- 7 + 11, 5 + 6 + 8, 5 + 6 + 12) or reduced (*S0*-3 + 12, 4 + 5 + 7) running efficiency. But for all of them, even though the average efficiency values of the have's and have-not's were statistically different, their value ranges overlapped noticeably (Figs. 4b and 5b), suggesting that either the independent recruitment of any single combination is insufficient to produce high efficiency, or that multiple combinations may lead to similarly high or low efficiency. We therefore asked whether a runner, over training, would have to both acquire and suppress a specific collection of different merging patterns to attain highly efficient running.

Given the similarity in the muscular composition and plausible biomechanical functions subserved by *S0*-5 + 6 + 8 and 5 + 6 + 12 (see Discussion), let us consider them together as a single group. Let us further denote *S0*-7 + 11 by E1 (E for enhancing), *S0*- 5 + 6 + 8/5 + 6 + 12 by E2, *S0*-3 + 12 by R1 (R for reducing), and *S0*-4 + 5 + 7 by R2. We systematically evaluated, over

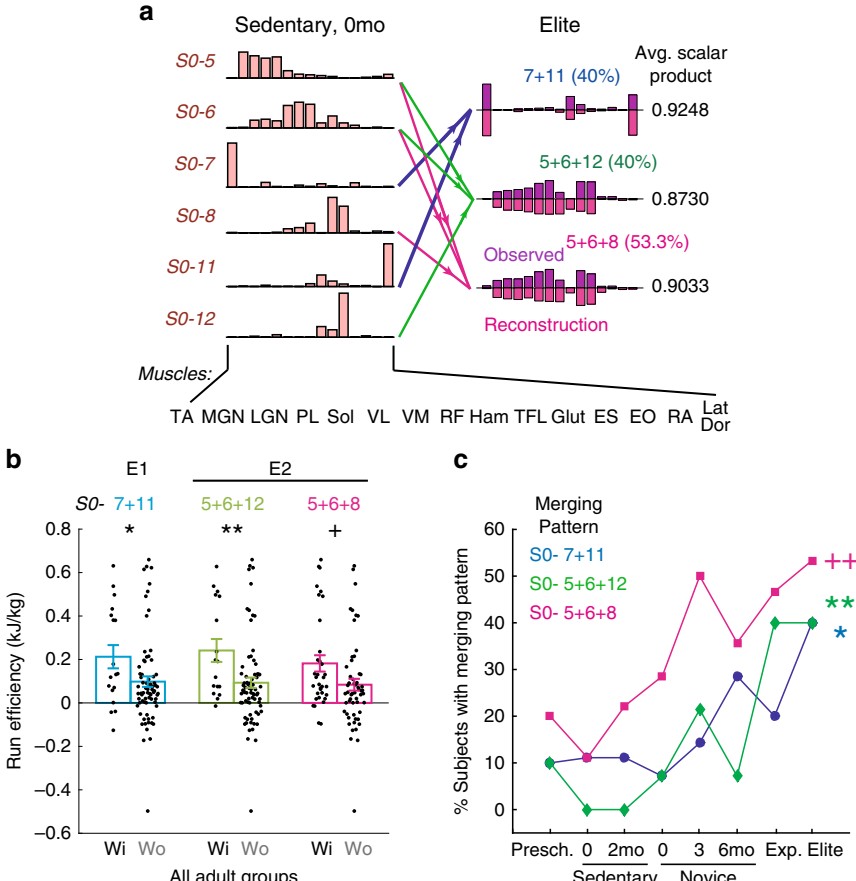

**Fig. 4 Muscle synergy merging patterns associated with increased running efficiency. a** Three efficiency-enhancing synergy patterns in adult runners (see also **b**) based on merging different combinations of Sedent0 synergies. Shown here for each merging combination are the Elite synergies being explained as merging (dark purple) and their reconstructions by merging their respective *S0-* combinations (light purple), averaged across all merging instances belonging to that combination. Arrows denote patterns of merging (magenta, $S0-5+6+8$, $n = 15$; blue, $7+11$, $n = 8$; green, $5+6+12$, $n = 8$). The percentage of Elite possessing each combination is shown in parentheses. **b** The merging combinations in **a** are potentially biomechanically relevant, in that across all adults, the subjects with (Wi) the merging pattern ($n = 18$, 17, 34 for $S0-7+11$, $5+6+12$, and $5+6+8$, respectively) had higher average running efficiency than those without (Wo) ($n = 72$, 73, 56) (mean ± SE; *$p = 0.049$; **$p = 0.011$; +$p = 0.028$; 2-tailed Mann–Whitney). **c** The frequencies of occurrence of the combinations in **a** increased from the groups with the least (Sedent0) to most (Elite) running experience (*$r$ across adults $= 0.83$, $p = 0.021$, 2-tailed t-test; **$r = 0.88$, $p = 0.0091$; ++$r = 0.90$, $p = 0.0058$), suggesting that their emergence may be related to running training. Source data for all panels are available as a Source Data file.

different combinations of E's and R's, how the presence of R1 or R2 impacted on the running efficiency of adults when either E1 and/or E2 were present or absent. For adults already possessing one or both E's, the presence of R1 or R2 brought the running efficiency to ~0 kJ kg$^{-1}$, but their absence increased efficiency to ≥0.2 kJ kg$^{-1}$ on average (Fig. 6a). For adults who did not acquire one or both E's, the presence of R1 or R2 likewise resulted in low running efficiency (~0−0.05 kJ kg$^{-1}$), but their absence only produced a relatively small efficiency increase in most subjects (~0.1 kJ kg$^{-1}$) that was statistically insignificant in seven of the eight comparisons examined (Fig. 6b). Thus, regardless of whether the E's are acquired, if an R is present running efficiency remains low; if neither E is acquired but the R's are suppressed, efficiency improves modestly, if at all; the highest running efficiency is achieved when the E's are present and the R's are absent simultaneously.

We next wondered whether runners ought to acquire any one or both of E1 and E2 to achieve the highest running efficiency. After excluding the runners with any R's, we found that the efficiency averages of subjects possessing just E1, just E2, and both E1 and E2 were not statistically different from each other (Fig. 6c, rightmost group). Thus, the E's may enable similarly

high running efficiency, and perhaps represent distinct efficiency-enhancing strategies acquired by different runners. This observation is consistent with the fact that, among all the efficiency comparisons in Fig. 6a, b, the pair with the smallest $p$ value was the one for subjects with either or both E present, between those with or without any R (Fig. 6a, pair 7 from left; $p = 0.0015$, 2-tailed Mann–Whitney).

We hasten to add that other yet-to-be-identified motor patterns in addition to the E's and R's are almost certainly at work in determining efficiency. A few subjects had with very high efficiency but without possessing E1 or E2 (Fig. 6b, pairs 5–7 from left), and another few E-possessing subjects had very low efficiency even when the R's were absent (Fig. 6a, pair 7 from left).

Plots of the merging patterns' frequencies of occurrence across subject groups suggest that acquisition of the E1-or-E2-without-R patterns results from prior running training. Across adults, prevalence increased significantly for both the E1-or-E2 (Fig. 6d) and E1-or-E2-without-R1 patterns (Fig. 6e), from 22% in Sedent0 to 73% in Elite. The latter correlation further suggests that activation of the E's and suppression of R1 may be acquired concurrently during training, consistent with the small number of

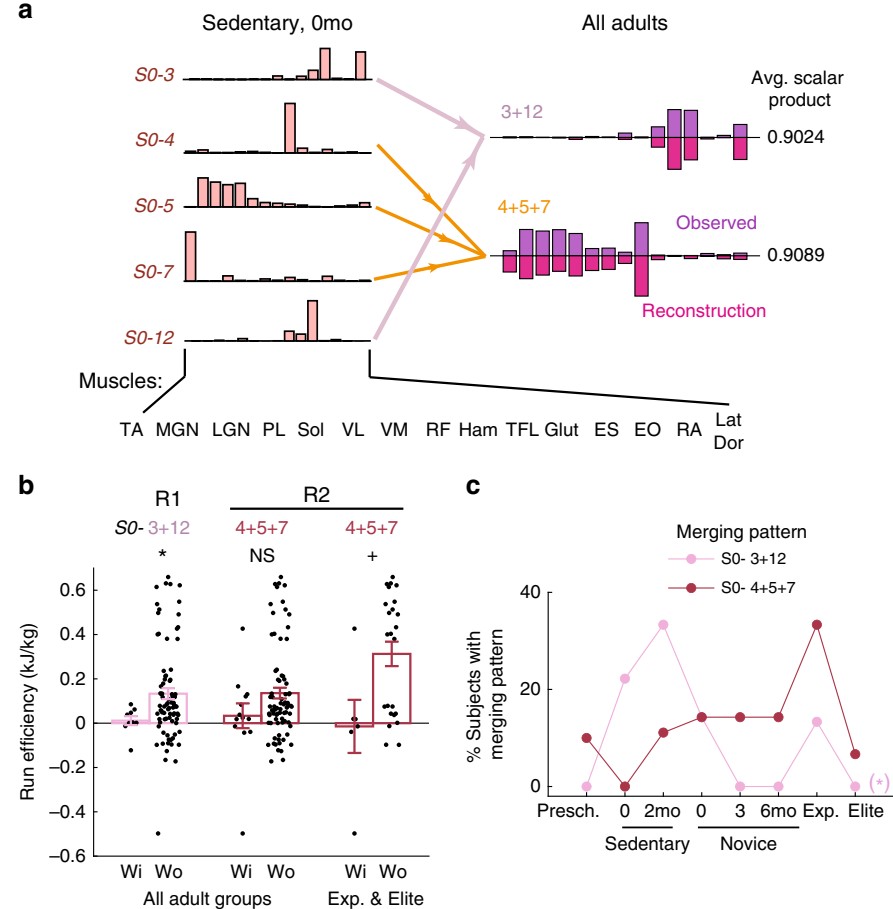

**Fig. 5 Muscle synergy merging patterns associated with decreased running efficiency. a** We identified two merging patterns of muscle synergies for which the adults having any of them had decreased running efficiency than those without it. To illustrate these patterns, for each combination we show the adult synergies being explained by merging (dark purple) and their reconstructions by merging their respective *S0-* combinations (light purple), averaged across all merging instances belonging to that combination. In both, the reconstructed vectors matched the original vectors very well (SP ≥ 0.90). Arrows denote patterns of merging (orange, $S0-4+5+7$, $n = 16$; light purple, $3+12$, $n = 9$). **b** For the merging combinations in **a**, the subjects with (Wi) the merging pattern ($n = 9, 13, 6$, for $S0-3+12$, $4+5+7$, and $4+5+7$ in Exp & Elite, respectively) had lower average running efficiency than those without (Wo) ($n = 81, 77, 24$) (mean ± SE; *$p = 0.042$; NS, $p = 0.24$; +$p = 0.016$; 2-tailed Mann–Whitney). **c** The frequency of occurrence (% subjects) across all subject groups for the merging combinations shown in **a**. Specifically, combination $S0-3+12$ was more prevalent in the less experienced groups (Sedent0/2) than the more experienced groups (Novice0/3/6, Exp, Elite) ((*), *r* over adults = −0.72, $p = 0.066$, 2-tailed *t*-test). Combination $S0-4+5+7$ was specifically more prevalent only in Exp. Source data for all panels are available as a Source Data file.

subjects found to possess both the E's and R1 (Fig. 6a). Interestingly, when subjects with the E's and without either R1 or R2 were considered, prevalence from Sedent0 to Elite generally increased but dropped prominently at Exp (Fig. 6f), consistent with R2's high frequency of occurrence only in Exp (Fig. 5c) and the unexpectedly low running efficiency in Exp (Supplementary Fig. 3B) (see Discussion).

**Synchronous drives to multiple synergies leading to merging.** We next sought to gain insight on a plausible mechanism of muscle-synergy merging by analyzing the temporal patterns ($C(t)$; Fig. 1) of the to-be-merged and merged synergies, assuming that a synergy's $C(t)$ represents the temporal activity of a neuronal oscillator that drives, from a higher layer, the last-order premotor interneurons that encode the synergy's muscle coordinative structure (**W**)[27]. If, over a running cycle, the $C(t)$ of the merged synergy resembles the $C(t)$ of one of the original synergies contributing to the merging (denoted by $C^*(t)$) but not those of the others, synergy merging results from the **W**-encoding networks of the original synergies being synchronized by drives from $C^*(t)$

while the other original $C(t)$'s cease to be active (Fig. 7c). If, on the other hand, all the original $C(t)$'s contribute equally to the final merged $C(t)$, synchronization of the **W**-encoding networks is achieved by a reconfigured network that oscillates with a temporal pattern that combines the original $C(t)$'s. The validity of these scenarios can be tested by linearly combining the to-be-merged $C(t)$'s to reconstruct the merged $C(t)$ in detected instances of synergy merging based on analysis of the **W**'s (Fig. 7a), and examining whether the scalar combination coefficient for one of the $C(t)$'s dominates over those for the others.

We performed the analysis described above on the 10 combinations of Sedent0 muscle synergies whose merging appeared in Elite (Supplementary Table 3). In seven of them, the scalar coefficient for one of the $C(t)$'s in the synergy combination was very noticeably larger than those for the others (largest coefficient = 78–99% of the sum of all coefficients; Fig. 7b). Thus, merging of muscle synergies during running training may be understood as a process that reassigns multiple original **W**-encoding networks to be driven by one of the original oscillators (Figs. 7c and 8).

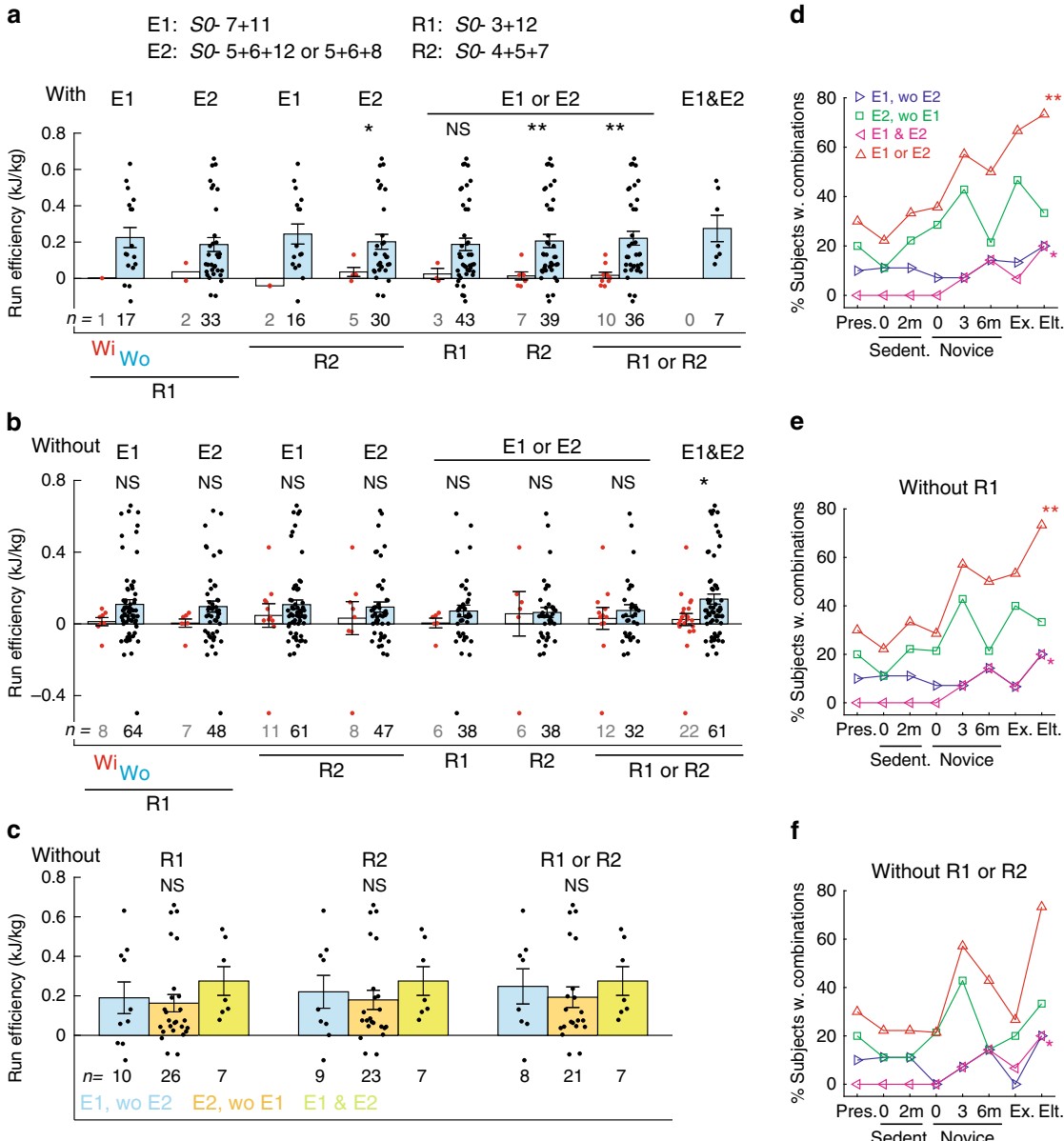

**Fig. 6 Acquiring and suppressing merging patterns for higher running efficiency. a** Running efficiency of different subsets of adults already possessing the efficiency-enhancing $S0$-$7 + 11$ (E1), $5 + 6 + 12$ or $5 + 6 + 8$ inclusive (E2), E1 or E2 inclusive, or E1 and E2, but with or without the efficiency-reducing $S0$-$3 + 12$ (R1) or $4 + 5 + 7$ (R2). In each of the eight pairs of bars (mean ± SE), the left denotes values in subjects with (Wi) the indicated R combination, and the right, subjects without (Wo) the combination. P values for the four sets with number of Wi subjects ≥3 were (left to right), $p = 0.040$, 0.17, 0.0071, 0.0015 (2-tailed Mann–Whiney; **$p < 0.01$; *$p < 0.05$; NS, $p > 0.05$). When E was present, the presence of R decreased the running efficiency. **b** Running efficiency of different subsets of adults already not possessing E1, E2, E1 or E2 inclusive, or E1 and E2, with or without R1 or R2. P values for the eight sets were (left to right) $p = 0.12$, 0.17, 0.85, 0.97, 0.27, 0.38, 0.86, 0.048 (2-tailed Mann–Whitney; *$p < 0.05$; NS, $p > 0.05$). When E was absent, the absence of R only increased efficiency modestly without statistical significance. **c** Running efficiency of adults without the R's, but with either E1 only (blue), E2 only (orange), or E1 and E2 (green). P values were (left to right) $p = 0.21$, 0.31, 0.38 (Kruskal–Wallis; NS, $p > 0.05$). The presence of any E combinations increased efficiency about equally when R was absent. **d–f** Prevalence (% subjects in the group with combination), across subject groups, of combinations E1 only (blue), E2 only (green), E1 and E2 (magenta), and E1 or E2 (red). Percentages were calculated without excluding subjects with R (**d**), after excluding those with R1 (**e**) or those with R1 or R2 inclusive (**f**) (** in **d**, r across adults = 0.96, $p = 4.5 \times 10^{-4}$, 2-tailed t-test; ** in **e**, $r = 0.91$, $p = 0.0039$; *$r = 0.86$, $p = 0.013$). Acquisition of E1 or E2 together with suppression of R1 correlated best with running experience (**). In **f**, the decrease of the E1-or-E2 frequency in Exp reflects the prominence of R2 – and hence the lower efficiency – in the group (Supplementary Fig. 3B). Source data for all panels are available as a Source Data file.

## Discussion

Overall, our data have provided a novel view of how the neural constraints on motor patterns may be reshaped for the motor system to adapt to the changing biomechanical and other constraints that arise during development and training. Developmental changes of the running motor patterns result in part from fractionations of muscle synergies (Fig. 3). We interpret this fractionation as a process that adapts the preschoolers' synergies to the variable, constantly changing neuro-musculoskeletal properties of the developing leg. Likely shaped by genetics, early experiences and environmental influences, the Presch synergies, with the many co-contractions involved (e.g., $S0$-$5 + 7$,

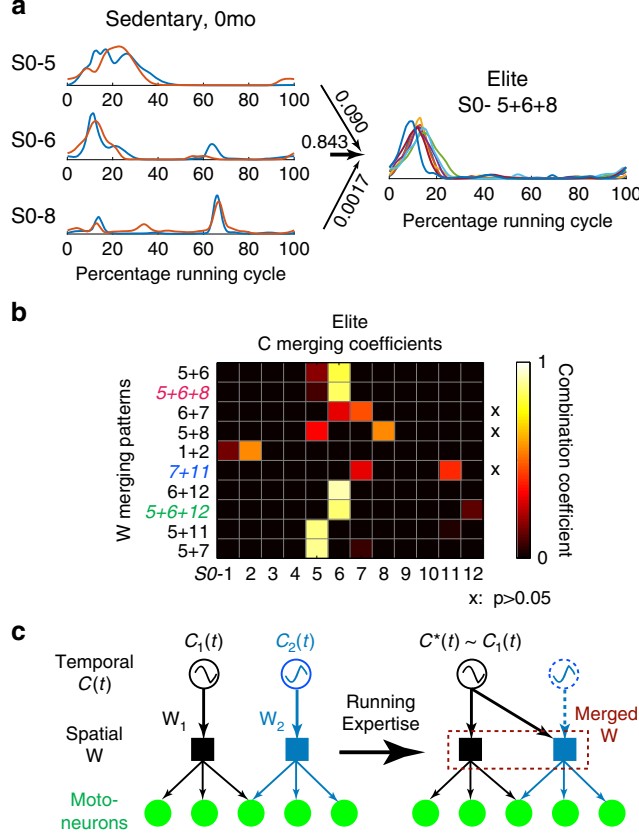

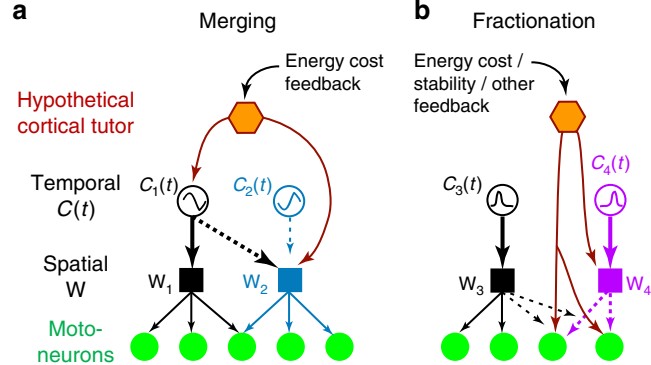

**Fig. 8 Hypothetical neural mechanisms of muscle-synergy merging and fractionation. a** We speculate that within the spinal cord, there exist multiple oscillators that generate burst activities at different phases of the gait cycle, and each oscillator can be assigned to provide the $C(t)$ of a flexible collection of downstream networks that encode the W's of the muscle synergies. In this example (from Fig. 7c), synergy merging amounts to reassigning $C_1(t)$ from driving just $W_1$ to both $W_1$ and $W_2$ while at the same time dissociating $C_2(t)$ from $W_2$. This reassignment can conceivably be directed by cortical tutoring inputs[53] that coactivate $C_1(t)$ and $W_2$ during training, and driven by sensory feedback related to energy cost, proprioception, and sense of effort[51]. **b** Fractionation of a muscle synergy during development can likewise be achieved by reassigning a network being driven by an idle oscillator ($C_4(t)$ in this example) to the motoneurons of a subset of muscles (the two on the right side) within an existing synergy ($W_3$). Descending inputs that co-activate the $W_4$-encoding network and the motoneurons of the fractionated muscles can reinforce these connections over time. Feedback related to gait stability and energy cost can drive this fractionation process.

**Fig. 7 Merging arising from synchronous drives to multiple synergies. a** In all detected instances of **W** synergy merging between Sedent0/Elite subject pairs, the $C(t)$'s of the to-be-merged (Sedent0) and merged (Elite) muscle synergies were resampled across time to produce cycle averages. The Sedent0 $C(t)$'s were then fit to a non-negative linear combination model to identify the combination coefficients that best explained the $C(t)$ of the merged, Elite synergy. Here, the $C(t)$ merging of $S0$-5, 6, and 8 from 2 Sedent0 subjects into $S0$-5 + 6 + 8 in eight Elite subjects ($n = 15$ instances of merging) is shown as an example. Merging coefficients for $S0$-5, 6, and 8 were 0.09 ± 0.11 (mean ± SD), 0.84 ± 0.11, and 0.0017 ± 0.0033, respectively. The coefficient of $S0$-6 obviously dominates this combination. **b** Shown here is the combination coefficients for the $C(t)$'s (per model in **a**) for the 10 most prevalent merging combinations in Elite (averaged across merging instances). In 7 of 10 merging combinations, the $C(t)$ of 1 to-be-merged synergy dominated the combination (largest combination coefficient = 78–99% of total coefficient values). In the remaining three (marked by x), the combination coefficients were not statistically significant across the to-be-merged synergies ($S0$-6 + 7, $p = 0.34$; 5 + 8, $p = 0.16$; 7 + 11, $p = 0.56$; 1-way ANOVA). Numbers of merging instances, from top to bottom, were, $n = 56, 15, 7, 8, 22, 8, 6, 8, 11, 6$. **c** A model that accounts for muscle-synergy merging by reassigning multiple synergy-encoding interneurons ($W_1$ and $W_2$) to be driven by the same oscillator for one of the original synergies ($C^*(t) = C_1(t)$), and with the original $C_2(t)$ ceasing to be active (blue dotted line). This model is consistent with results in **b**. Source data for **a**, **b** are available as a Source Data file.

with plantar- and dorsi-flexors; Supplementary Table 2, pattern 1), may be required to ensure stability in running[28] given the immature neural control of posture and balance at that age[29,30]. As the child grows, the need of such co-contractions for stability steadily diminishes. This permits the gradual maturation of anthropometries and other neuro-musculoskeletal properties to provide the sensory signals necessary for triggering individuated control of muscle subgroups within the original synergies, so that running stably with the adult legs can now be achieved. Other

mechanisms are likely involved in this development, including the elimination of some Presch synergies (e.g., cluster $P$-7 in Fig. 3a) that may no longer be suitable for adult running. For a development untinged by training, gradual fractionations of muscle synergies could be a maturation process that generates control flexibility so that running can be performed under a variety of circumstances[2,31] despite variations of plant properties during development, even though after fractionation running may not be most efficiently executed. Interestingly, synergy fractionation has very recently been demonstrated during the early development of stepping[32], suggesting that fractionation may contribute to the developmental reorganization of motor patterns for multiple modes of locomotion.

After the plant properties stabilize, to run more efficiently training is required to sculpt new motor patterns. Training-related changes of the running motor patterns result from merging of muscle synergies (Fig. 4) underscored by reassigning the to-be-merged synergies to be synchronized by one of the original oscillators (Fig. 7). In essence, learning to merge specific muscle synergies amounts to discovering the subspace of motor commands, within the larger space defined by the original pre-training synergies, that can produce running forms with more efficiency. After merging, not only is the number of degrees of freedom to be specified reduced, but motor-pattern variability is also restricted within the space of commands relevant to the efficiency level demanded of elite runners. Muscle-synergy merging is equivalent to reducing task-relevant motor variability through learning[33].

How may the merging of specific synergies lead to increased running efficiency? We note that two of the three efficiency-enhancing merging combinations, $S0$-5 + 6 + 8 and $S0$-5 + 6 + 12 (E2), involved co-activations of the lower-limb extensors

(ankle plantar-flexors in S0-5, quadriceps in S0-6, and gluteus maximus in S0-8/12) and TFL (in S0-8/12) during stance (Supplementary Fig. 5). The TFL, especially its portion with oblique fibers, is an important stabilizer of hip extension; these merging combinations may thus function to enhance propulsion during late stance of the running cycle[34]. Indeed, increased muscle co-activations were observed in elite Kenyan runners, but not in other runners who ran at higher energetic costs in most testing speeds[35]. The other efficiency-enhancing combination, S0-7 + 11 (E1), involved co-activations of latissimus dorsi (S0-11) and the ankle dorsiflexors (S0-7). This combination may enable functional synchronizations of arm- and leg-swing during running. As natural arm swing counters vertical angular momentum of the lower limbs[36] and minimizes head, shoulder and trunk rotations[37], such synchronizations may reduce energy loss and improve economy. Indeed, arm swing contributes to vertical and forward acceleration when running speed increases[34].

In addition, we have identified synergy merging combinations whose presence tended to increase energy loss during running (Figs. 5 and 6). Combination S0-4 + 5 + 7 (R2) involved co-contraction of ankle plantar- and dorsi-flexors at the initial contact of the running cycle (Supplementary Fig. 5), which would increase both the ankle joint stiffness[38] and vertical loading rate[39], thus increasing energy loss[40]. Combination S0-3 + 12 (R1) involved simultaneous activations of trunk muscles and gluteus maximus at early stance and early swing (Supplementary Fig. 5). We speculate that it increases energy loss by eliciting unnatural, inefficient arm swing[41]. Indeed, we found only 1 adult who possessed both R1 and E1, the other arm-swing related combination (Fig. 6a, leftmost pair), suggesting that they may reflect two alternative strategies of arm swing, one being more inefficient than the other.

Our results imply that screening of muscle synergies in runners may facilitate the spotting of efficiency-reducing running strategies that are otherwise not identifiable just by visual evaluation of the running form and posture. This may be especially so for experienced runners who already run with mature forms. For instance, the Exp subjects here ran with surprisingly low efficiency (Supplementary Fig. 3B), which may be related to the high prevalence of S0-4 + 5 + 7 (R2) in this group (Fig. 5C). For these subjects, this combination probably reflects an unrecognized, inadvertently acquired bottleneck that they must overcome before attaining the Elite's efficiency.

We have correlated muscle-synergy changes between groups with either age-related development or training-induced improvement in running efficiency. Since both age and training experience correlate with the preferred running speed (Supplementary Fig. 6A)[42], one may interpret that all synergy changes observed here are primarily driven by the across-group variation of running speed. When our three measures of synergy changes – the number of muscle synergies, synergy-vector sparseness, and the Merging Index – were correlated with body height-normalized preferred running speed, not surprisingly the correlations were either significant or nearly significant (Supplementary Fig. 6B–D), but their correlation coefficients were low ($|r| = 0.20$–$0.36$). When the correlations were performed within each subject group, none was significant ($p = 0.15$–$0.97$, 2-tailed $t$-test). Thus, either the running speed is not the sole determinant of synergy changes, or synergy changes are related to other variables that loosely correlate with speed.

We argue that fractionation and merging of muscle synergies are processes primarily driven by development and training, respectively. Not only were the merging combinations not the exact opposite of the fractionation patterns (which would be expected if synergy changes across Presch and adults were just speed-driven) (Supplementary Table 2, 3; Supplementary Note 5,

Supplementary Fig. 7), but the presence or absence of the five energetically relevant merging patterns also separated the high- and low-efficiency subjects across groups with different preferred speeds. Indeed, for three of the five patterns, the preferred running speeds of the have's and have-not's were not statistically different (Supplementary Fig. 6E). Most importantly, longitudinal changes of the three measures of synergy changes (Fig. 2) were noted across the time points of the Novice subjects, each of whom ran at the same speed at all points.

Across our subject groups, the number of muscle synergies deployed tended to decrease as the preferred running speed increased (Supplementary Fig. 6B). Interestingly, Yokoyama et al.[11,43]. observed that in male runners and non-runners, the number of synergies tended to increase as the running speed increased. Our finding is not necessarily inconsistent with this result. It is plausible that when running at the self-selected preferred speed, one employs a set of designated, default-mode synergies whose dimensionality decreases with higher preferred speeds and training; but when running at speeds beyond the preferred, additional synergies are recruited. Indeed, Yokoyama et al.[11] tested each subject over his full speed range ($\sim$9–16 km h$^{-1}$) while we tested ours at their preferred speeds (5–12 km h$^{-1}$), also likely to be at the lower end of their natural speed ranges. Further experiments are needed to clarify these dependencies.

Our analysis of the synergies' temporal coefficients suggests that muscle-synergy merging may result from reassigning the $\mathbf{W}$'s of multiple synergies to an oscillator that drives one of the original $C(t)$'s (Fig. 7c). Recent data from multiple model vertebrates have shown convincingly that locomotor $\mathbf{W}$'s and $C(t)$'s are encoded by spinal interneuronal networks[8,19,44,45]. Results from human survivors of stroke[46,47] and spinal cord injury[48] also partially support the subcortical origin of the locomotor synergies. Thus, the observed $\mathbf{C}$-to-$\mathbf{W}$ reassignments for merging synergies are presumably underscored by plastic reorganization of the synergy-encoding subcortical networks.

We speculate that within the spinal cord and/or brainstem, there exist multiple oscillators that can generate rhythmic activity bursts at different phases of the running cycle, and that each of them can be flexibly assigned to drive any collection of the downstream $\mathbf{W}$'s. This reassignment can be directed by supraspinal (e.g., motor cortical) inputs as these inputs synchronously excite the oscillator and the $\mathbf{W}$'s to be merged under it (Fig. 8a). During early training, such synchronous activations may occur by chance as the runner explores the motor-command space through motor variability;[49,50] subsequently, it can be reinforced by reward signals or feedback sources that signify a reduction in energy expenditure[51,52]. This hypothetical neural mechanism of muscle-synergy merging echoes the previous findings that the motor cortex tutors subcortical circuits by guiding their plastic rearrangements during the learning of non-dexterous skills[53], and that changes in motor cortical firing correlate with muscle-synergy changes during early learning[12]. Fractionation of a muscle synergy during development can be analogously achieved through supraspinally-directed synchronization of an oscillator-driven interneuron and the motoneurons of a subset of muscles within a to-be-fractionated synergy (Fig. 8b).

Overall, our results have provided a framework for understanding how the outcome of motor development constrains adult motor learning. In our data, some of the fractionated muscle synergies from development were merged during training, albeit in merging combinations that were different from the fractionation patterns that derived them (e.g., P-6 was split into S0-6, -7 and -8, but, S0-6 and -8 were merged together with S0-5 instead of S0-7 in Elite; Supplementary Note 5). The fractionated synergies at the end of child-to-adult development are the

building blocks for subsequent training-induced recombination that allows novel motor patterns for accomplishing the learned behavior to emerge.

Our finding of training-induced synergy merging implies that to learn a new skill, the CNS exploits and modifies existing patterns at its disposal rather than assembling new patterns de novo. In Yokoyama et al.[11], both non-runners and runners select and perhaps modify different subsets of the same pool of synergies (see Supplementary Note 6). Ballet dancers, over years of training, tune their walking synergies so that the same synergies are also employed for difficult balance tasks[13]. Here, by comparing the synergies from runners with a wide range of training experience, we argue further that muscle-synergy merging can be one general mechanism that the CNS employs to generate new patterns from pre-existing synergies, and situate this mechanism as one that appears opposite to the process that drives developmental synergy changes. Interestingly, when merging happens to the inappropriate combinations, task performance can be degraded (e.g., S0-4 + 5 + 7 in Exp); thus, our concept of merging should also be useful in elucidating how training fails.

Our view implies that any factors during development that influence fractionation – from early exposure to specific physical activities, neuro-musculoskeletal injury, to nutrition – may facilitate or impede adult motor learning by biasing the chance for the CNS to discover the best merging combinations. We do not yet know whether developmentally-driven fractionation and training-driven merging may interact if running training begins before adulthood, and how this interaction may impact on the success of subsequent adult training.

## Methods

**Subjects**. Five groups of subjects – preschoolers; sedentary adults; and novice, experienced, and elite adult runners – were studied. From preschoolers to elite runners, they represent groups with progressively more running experience. The preschoolers and sedentary adults were recruited through online advertisements, and the adult runners, from various local running clubs. All enrolled preschoolers ($n = 10$; age = $4.2 \pm 1.6$ [mean ± SD] years old; four males, six females) were able to run with a flight phase during which both feet were not in contact with the ground. All enrolled sedentary adults ($n = 9$; age = $30.2 \pm 4.8$ years old; four males, five females) had not received any prior running training, and did not run or exercise regularly at the time of enrollment. They were also explicitly instructed to not partake in any substantial running activity during the time interval between their two recording sessions (see below). The novice runners ($n = 14$; age = $45.3 \pm 6.8$ years old; six males, eight females) had <3 months of prior running experience, and were about the start running training at the time of first recording (see below). The experienced runners ($n = 15$; age = $43.0 \pm 11.3$ years old; eight males, seven females) had 2–10 years of training in running ($4.57 \pm 2.09$ years). The elite runners ($n = 15$; age = $37.3 \pm 6.4$ years old; all male) had 3–30 years of previous training ($8.97 \pm 6.24$ years), and could all complete a full marathon within 3 h in the recent 2 years, proven with official documents issued by marathon organizing bodies. All subjects had not had any musculoskeletal injury for ≥1 year, and had no history of any neurological impairment. We ensured that all enrolled runners were not feeling exhausted from previous trainings or competitions at the time of recording, as exhaustion and fatigue could potentially influence kinematics and muscle activities. For all subjects, the right leg was the dominant leg.

All procedures were approved by the Departmental Research Committee of the Department of Rehabilitation Sciences, Hong Kong Polytechnic University (reference number: HSEARS20150730002). The parents or guardians of all preschoolers and all adult subjects gave informed consent before experimentation. The study design and conduct complied with all relevant regulations regarding the use of human study participants.

**Behavioral task**. All preschoolers ran over-ground at self-selected speed without assistance over a single session. Treadmill running was not possible for preschoolers due to safety considerations. Each preschooler ran in their own shoes on a straight, flat runway with force plates (model DBCEEWI, AMTI, Watertown, MA, USA) embedded midway of the path. But the quality and quantity of these force data were deemed insufficient for reliable estimates of ground reaction force, and thus they were not used. For all preschooler trials, the kinematics recorded (see below) were carefully inspected offline, so that all trials in which the subject was not running were discarded. For this group, a self-selected running speed of $7.4 \pm 2.3$ km h$^{-1}$ ($2.0 \pm 0.5$ body height s$^{-1}$ (B.H. s$^{-1}$)) was achieved.

All adult subjects were asked to run on an instrumented treadmill (Tandem treadmill of AMTI) in their usual running shoes at self-selected preferred speed for 2 min. Before data recording, subjects were given sufficient time for warm up and treadmill adaptation. The treadmill speed was gradually increased in 0.5-km h$^{-1}$ increments until it reached the subject's preferred speed. For the different groups, the speeds achieved were: sedentary (both sessions, see below): $6.2 \pm 0.9$ km h$^{-1}$ ($1.0 \pm 0.1$ B.H. s$^{-1}$); novice (all three sessions, see below): $6.8 \pm 0.9$ km h$^{-1}$ ($1.1 \pm 0.1$ B.H. s$^{-1}$); experienced: $7.5 \pm 1.3$ km h$^{-1}$ ($1.2 \pm 0.2$ B.H. s$^{-1}$); elite: $12 \pm 0$ km h$^{-1}$ ($1.93 \pm 0.05$ B.H. s$^{-1}$). We inspected the vertical ground reaction force data to ensure the presence of flight phase in each running trial. Subjects were allowed to rest upon request. In each session, data recording began only after the runner had accustomed to running on the treadmill, as indicated by visually stable running kinematics.

The same novice runners were followed up two times, at 3 and 6 months after the initial recording session and into their training, respectively, so that any biomechanical and EMG changes over their training course could be monitored. They were asked to enroll in training programs of their choices before recruitment, and did not receive any specific instruction on training from the experimenters. The sedentary subjects were followed up at 2 months after the initial session, and remained sedentary over this interval. Importantly, for all subjects with longitudinal sessions, the follow-up sessions were conducted at the same treadmill speed as that used in the first session. Experienced and elite runners were not followed up after the initial session.

**Data recordings**. In each session, bilateral lower-limb and trunk kinematic data during running were collected with an 8- (or, for preschoolers, 10-) camera motion capture system (Vicon MX, Oxford Metrics Group; Oxford, UK) at 200 Hz. All motion data were acquired using the software Vicon Nexus (version 1.8.5 for treadmill trials, 2.4.0 for overground trials; Oxford, UK). Ground reaction force during running was recorded at 1,000 Hz by force plates; for adults, the force plates were installed under the instrumented treadmill.

To assess muscle activities during running, wireless electrodes (Trigno, Delsys; Natick, MA, USA) were attached to skin surface to record electromyographic (EMG) signals at 1,000 Hz. Acquisition of EMGs was achieved using the software Trigno$^{TM}$ Control Utility (version 2.6.12; Delsys; Natick, MA, USA). Recordings were obtained from 15 trunk and lower limb muscles on the right side, including: latissimus dorsi (LatDor), external oblique (EO), rectus abdominalis (RA), erector spinae (ES), gluteus maximus (GLUT), tensor fasciae latae (TFL), vastus medialis (VM), vastus lateralis (VL), rectus femoris (RF), bicep femoris (HAM), tibialis anterior (TA), medial gastrocnemius (MGN), lateral gastrocnemius (LGN), soleus (SOL) and peroneus longus (PL). Placement positions of the electrodes were identified using guidelines from the Non-Invasive Assessment of Muscles-European Community Project (SENIAM) (www.seniam.org) whenever possible. Before electrode placement, skin surface was cleaned with alcohol wipes, and excess body hair was shaved. Electrodes were then fixed onto skin with double-sided tape. To reduce motion artifact during recording, positions of all electrodes on the limbs were mechanically stabilized by wrapping self-adherent dressing (3M$^{TM}$ Coban$^{TM}$) around the thigh and crus. Care was taken to ensure that the electrodes and wrapping did not obstruct the subjects' movement in any way. All force and EMG data recorded were synchronized.

**Biomechanical analysis**. To assess whether changes in neural control may be related to running performance optimization, we performed biomechanical analysis to correlate changes in EMG-derived muscle synergies (see below) to biomechanical parameters. Ground reaction force affects running performance significantly because of its relation with energy loss and body-stiffness modulation, both of which can be derived from the vertical ground reaction force (VGRF). The VGRF data collected from our force plates were low-pass filtered (4th order Butterworth; cutoff of 50 Hz), and we calculated the vertical center of mass (CoM) acceleration based on the body mass and the VGRF. The vertical CoM displacement was calculated by integrating the vertical CoM acceleration twice[54]. To obtain the constant for the first integration, we further assumed nil average velocity of CoM over each stride[54,55]. To characterize the running energetics of every subject, a hysteresis loop was formed by plotting VGRF against CoM displacement during stance; the vertical-direction energy loss was represented by the difference between the negative CoM work and the positive CoM work, normalized by body mass[56] (Fig. 1c). For every subject, we specifically computed the vertical-direction energy loss per kg body mass over 30 min of running through an estimation of the average energy loss per running cycle.

Within the data from the sedentary (both sessions) and experienced adult runners, we noticed a positive, statistically significant correlation ($r = 0.443$, $p = 0.0125$, 2-tailed $t$-test) between preferred body-height normalized running speed and the 30-min energy loss. For the elite and novice subject groups, their energy loss values tended to lie below the regression line that described the data of sedentary and experienced adults (Supplementary Note 3, Supplementary Fig. 3). We, therefore, inferred that each subject's running efficiency (in kJ kg$^{-1}$) may be approximated by how much the energy loss value lies vertically below this regression line, which denotes the expected energy loss for the less well-performed runners given a preferred running speed. With this definition, the running efficiency was then compared across all adult groups.

**EMG pre-processing.** All EMGs collected were analyzed using custom functions written in Matlab (R2016b and R2019b; Mathworks; Natick, MA, USA). Raw EMGs were first high-pass filtered (Finite Impulse Response filter, or FIR; cutoff at 50 Hz), then rectified, and then low-pass filtered (FIR, cutoff at 20 Hz)[57]. The data points were then integrated at 20-ms intervals. Occasional high-amplitude spikes arising from motion artifact or noise were removed by visual examination of the recordings. After filtering, the EMG amplitude of every muscle was normalized to unit variance[58].

**Muscle synergy analysis: an overview.** Muscle synergy analysis seeks to uncover, from the EMGs, muscle coordinative structures that are utilized as neuromotor control units[59]. It is assumed that these structures, or muscle synergies, are reflected by the statistical regularities of muscle co-activations embedded within the variability of EMGs[60]. Mathematically, muscle synergy may be defined in various ways[61]. Here, each muscle synergy is a time-invariant unit that co-activates multiple muscles according to a balance profile, and activated by a temporal waveform (Fig. 1b)[57,58]. In our analysis, we first identified the muscle synergies of each subject by applying a factorization algorithm to the EMGs. We then proceeded to characterize how the synergies varied across groups, and revealed specific changes of the synergies that are dependent on either development or the stages of running training. To understand the functional significance of these alterations, for the adult groups we further correlated some specific synergy changes to improvement in running efficiency. Lastly, we analyzed the synergies' temporal activations, examined their changes *vis-à-vis* across-group changes of the muscle synergies, and argued how our description of the temporal activations sheds light on the mechanism driving the changes of the muscle synergies as subjects trained on running.

**Identification of muscle synergies.** Muscle synergies were extracted from the EMGs of each session of each subject using the Non-negative Matrix Factorization algorithm (NMF), which decomposes muscle activities ($\mathbf{D}(t)$) into a linear combination of time-invariant synergy vectors ($\mathbf{W}_i$) scaled by time-varying activation coefficients ($C_i(t)$) through a set of iterative multiplicative update rules[62]. Thus, the EMGs can be reconstructed according to the following equation,

$$\mathbf{D}(t) \approx \sum_{i=1}^{N_{syn}} C_i(t)\mathbf{W}_i. \tag{1}$$

To determine the number of muscle synergies, $N_{syn}$, NMF was applied to successively extract 1, 2, …, 15 muscle synergies from the data of each session. A maximum of 15 synergies were extracted because 15 muscles were simultaneously recorded. The appropriate number of muscle synergies was determined as the minimum number required for an EMG-reconstruction $R^2$ of ~80%[26]. Following the general definition of $R^2$ suitable for use with the NMF[63], the $R^2$ was calculated as follows,

$$R^2 = 1 - \frac{SSE}{SST}; \tag{2}$$

$$SST = \sum_{i,j}(\mathbf{D}_{ij} - mD_i)^2, \quad SSE = \sum_{i,j}(\mathbf{D}_{ij} - [\mathbf{WC}]_{ij})^2; \tag{3}$$

where SST is the sum squared total, $\mathbf{D}_{ij}$ is the EMG data of the $i$th muscle at the $j$th time point, $mD_i$ is the average EMG value of the $i$th muscle, and SSE is the sum squared error. To prevent the extracted synergy set from representing a suboptimal local minimum on the error surface, each instance of synergy extraction was repeated 20 times, each time with $\mathbf{W}_i$ and $C_i(t)$ initiated with different uniformly distributed random values drawn from the open interval between 0 and the maximum EMG value. The solution yielding the highest $R^2$ was selected for downstream analysis. For every NMF implementation, the update rules were terminated when a between-iteration change of EMG-reconstruction $R^2 < 0.001\%$ was observed in 20 consecutive iterations[58].

**Clustering muscle synergies.** To characterize how the muscle-synergy vectors differed between subject groups, we first identified the representative synergy vectors in each group by $k$-means clustering. This clustering was performed using the Matlab function $k$-means (Statistics toolbox), implemented with the squared-Euclidean metric, and with the initial cluster centroid positions chosen uniformly randomly from the to-be-clustered synergy vectors. Each clustering was repeated 1000 times with different initial centroid estimates; the replicate with the smallest point-to-centroid sum was chosen.

The number of synergy clusters in each subject group was identified by computing the gap statistic[64], which measures the compactness of the clustering achieved against those in reference data sets without any obvious clustering. Reference data sets ($N = 500$) were first created by sampling uniformly from within the bounds of the original muscle-synergy set; each of them was then clustered by $k$-means (100 replicates), at 2–20 clusters. The optimal number of clusters was then the smallest number, $k$, such that

$$Gap(k) \geq Gap(k+1) - sd(k+1), \tag{4}$$

where $Gap(k)$ is the gap statistic at $k$ clusters, and $sd(k)$ is the standard deviation of the clustering compactness in the reference data sets[64].

**Muscle synergy sparseness.** When we inspected the muscle-synergy vectors, it appeared to us that the number of active muscle components within the synergy vectors varied systematically across subject groups. We, therefore, quantified the sparseness ($\varphi$) of each muscle synergy with the following definition[65]:

$$\varphi = \frac{\sqrt{n} - \frac{\sum_{i=1}^{n}|\mathbf{W}_i|}{\sqrt{\sum_{i=1}^{n}\mathbf{W}_i^2}}}{\sqrt{n} - 1}, \tag{5}$$

where $\mathbf{W}_i$ is the $i$th muscle component of the $\mathbf{W}$ synergy vector, and $n = 15$ is the number of muscles in the vector. Per this definition, a very sparse vector with a single non-zero component has $\varphi = 1$; a non-sparse vector with equal components across all muscles has $\varphi = 0$. For every subject, an average $\varphi$ was calculated across the subject's muscle synergies; across groups, the group averages of the subjects' $\varphi$'s were compared.

**Muscle synergy similarity.** Similarity between the muscle synergies of two subject groups was first quantified by the scalar product between the centroids of the synergy clusters (normalized to unit vectors). For every comparison, each of the relatively subject-invariant clusters (defined as having synergies from ≥1/3 of the subjects) of a group was matched to a cluster in another group by maximizing the total scalar product values in the matching. The synergy clusters that could not be matched with scalar product ≥0.8 were classified as unmatched. For every matched pair of synergy clusters, the weightings of each muscle within the muscle-synergy vector were then independently compared between the two groups.

The generalizability of the muscle synergies from each subject group to describing the EMGs of another group was evaluated by cross-fitting. The synergy matrix, $\mathbf{W}$, of each subject from one group was fit to the EMGs of every subject in another group. This fit was accomplished by the NMF algorithm, with $\mathbf{W}$ held constant while $\mathbf{C}$ was updated across iterations. The quality of the cross-fit was quantified by the $R^2$ for reconstructing the EMGs with the constant $\mathbf{W}$ and the updated $\mathbf{C}$. As a benchmark for comparison, we then fit the synergies of each subject to the EMGs of other subjects in the same group. For both the across-group and within-group fits, an average $R^2$ was obtained across subject pairs. The difference between the across- and within-group cross-fit $R^2$ values essentially indicates the extent to which the synergies of one group may generalize to describing the data of another; the more negative the $R^2$ difference, the less able can the synergies of one group generalize to the group being fit (Supplementary Fig. 2).

**Merging of muscle synergies.** After we computed and compared the sparseness of the muscle-synergy vectors in all subject groups, it appeared that the more experienced in running a subject group is, the less sparse the group's synergies are. Careful visual inspection of the muscle-synergy weightings in the different groups of runners showed that the synergies from the more experienced groups may be the result of combining the weightings (or merging[26,47]) of specific synergies in less experienced groups. To systematically explore how the degree of muscle-synergy merging changes as runners train, we modeled the merged synergy as a linear combination of the contributing synergies[26]:

$$\vec{w}_i \approx \sum_{k=1}^{N^b} m_k^i \vec{w}_k^b, \quad m_k^i \geq 0, \quad k = 1\ldots N^b \tag{6}$$

where $\vec{w}_i$ is the $i$th merged muscle synergy, $\vec{w}_k^b$ is the $k$th synergy to be merged, $N^b$ is the number of synergies contributing to the merging, and $m_k^i$ is a non-negative coefficient that scales the $k$th synergy in the merging. The coefficient $m_k^i$ was calculated through a non-negative least squares fit, implemented using Matlab (function lsqnonneg) after $\vec{w}_i$ and $\vec{w}_k^b$ were normalized to unit vectors. Following[26], an instance of synergy merging was identified when $N^b \geq 2$, $m_k^i \geq 0.2$ for all $k$, and the scalar product between $\sum_{k=1}^{N^b} m_k^i \vec{w}_k^b$ and $\vec{w}_i$ was ≥0.8.

To assess whether the synergies of one subject group (say, group "A") may be explained as merging of specific synergies from another subject group (group "B"), we first identified the synergy cluster centroids of both groups (see above), and reconstructed each cluster centroid in "A" by merging the cluster centroids of "B". Instances of merging were then detected by noting the well-reconstructed centroids. While simple and intuitive, this assessment does not consider how within-cluster variability of the synergy vectors may affect merging. Thus, we also assessed between-group synergy merging with a more precise method. For every subject in "A", we reconstructed each synergy of the subject by combining the synergies of each subject in "B". We performed this reconstruction for every "A-B" subject pair, and compiled individual instances of merging across all pairs. These instances of merging were then classified according to the cluster membership of the "B" synergies contributing to the merging (e.g., all instances involving merging two synergies from clusters 5 and 6 of "B", respectively, are grouped together). This compilation then resulted in a list of merging combination of synergy clusters with variable number of contributing clusters. The importance of each combination was

then ranked by finding the percentage of "A" subjects in whose synergies that combination of merging was observed.

When assessing muscle-synergy merging using the second method above, the extent that the "A" synergies can be reconstructed by merging the "B" synergies can be conveniently quantified by the percentage of synergies of each "A" subject that are explained as merging. Here, this percentage is referred to as the Merging Index (MI). After "B" was fixed to a specific group, MI was calculated for every subject in all groups; averages of MI were then compared across subject groups.

**Correlating muscle synergies to biomechanics.** One goal of our muscle-synergy merging analysis is to identify specific training-dependent synergy merging combinations whose emergence correlates with running performance. To this end, we first identified all instances of synergy merging in all adult groups using the synergies of the least experienced Sedent0 subjects as the basis vectors in the merging combination. Among all merging instances, we then identified all unique merging combinations present, and shortlisted a subset of them with a prevalence of >30% in at least 1 subject group. For each combination in this select list, we then evaluated its potential contribution to performance by noting, in every of the seven adult groups, the subjects who possessed a muscle synergy that could be explained as a merging of that combination (with the synergies contributing to this merging coming from at least one Sedent0 subject). Across all seven adult groups, the running efficiency values of the subjects possessing this merging combination were then compared with those of the subjects without this merging combination. Muscle-synergy merging combinations whose presence or absence contributed to better running efficiency across adult groups were then identified.

For every Sedent0 cluster combination whose merging could be related to improved running efficiency, we also calculated the percentage of subjects possessing that merging combination in every of the eight subject groups. If the emergence of that merging combination is dependent on running training, an increase in this percentage from the least to most experienced groups is expected.

**Fractionation of muscle synergies.** When comparing the synergy sparseness values of the sedentary adult groups with other groups (the preschoolers in particular), we noticed that those of the former were considerably higher than those of the latter. This prompted us to explore whether the muscle synergies of the sedentary adults may result from fractionating specific preschooler muscle synergies. We modeled muscle-synergy fractionation as the converse of synergy merging:

$$\vec{\mathbf{w}}_i^{b} \approx \sum_{k=1}^{N^f} m_k^i \vec{\mathbf{w}}_k^{f}, \; m_k^i \geq 0, \; k = 1 \ldots \; N^f \tag{7}$$

where $\vec{\mathbf{w}}_i^{b}$ is the synergy to be fractionated, $\vec{\mathbf{w}}_k^{f}$ is the $k$th synergy fraction resulting from the split, $N^f$ is the number of synergy fractions, and $m_k^i$ is a non-negative coefficient. Likewise, the coefficient $m_k^i$ was calculated using non-negative least squares fit. Detection of instances of fractionation, identification of the fractionated muscle-synergy clusters of the preschoolers, and identification of the sedentary clusters representing the resulting fractions were all performed with procedures analogous to the ones described for merging.

**Analysis of synergy activation coefficients.** After detecting an instance of muscle-synergy merging based on analysis of **W**, we further explored whether the temporal activations (**C**) of the to-be-merged synergies could likewise be combined to account for the **C** of the merged synergy (Fig. 7a). We first segmented **C** of each synergy so that the segment boundaries corresponded to heel-strike times of the running cycles. The **C** segments were then resampled into 1000 time points per segment (Matlab function interp1; linear option), and averaged across cycles. This averaged **C** of each synergy, as a 1000-tuple, was then normalized to unit vector for downstream analysis.

For each instance of merging, the **C** of the merged synergy, $C^m(t)$, is modeled as follows:

$$C^m(t) = \sum_{k=1}^{M} s_k C_k^{tm}(t), \; s_k \geq 0, \tag{8}$$

where $C_k^{tm}(t)$ is the temporal activation of the $k$th to-be-merged muscle synergies, $M$ is the number of synergies contributing to the merging, and $s_k$ is the scaling coefficient for the $k$th to-be-merged synergy in this combination. As before, the $s_k$'s were identified by non-negative least squares. We were interested in testing whether the $s_k$'s of the $M$ synergies were approximately the same in magnitude, or whether the $s_k$ of one of them dominated this combination. The former scenario would suggest that merging of muscle synergies (the **W**'s) happens when the **W**-encoding networks are synchronized by a new, reconfigured network that oscillates with a temporal pattern that combines the original $C_k^{tm}(t)$'s. The latter scenario would imply that muscle-synergy merging happens when the **C** of one dominating synergy drives the temporal activations of all to-be-merged synergies while the other non-dominating $C(t)$'s cease to be active (Fig. 7c). For every biomechanically relevant merging combination identified, we, therefore, tested the null hypothesis that the $M$ $s_k$ values have the same mean using ANOVA or the Kruskal–Wallis test. For the ones with significantly different means, we calculated the largest $s_k$ as a percentage of the sum of all $s_k$ values. A high percentage implies the domination of one specific $C_k^{tm}(t)$.

**Statistics.** To evaluate whether samples from any two subject groups have a difference in mean or median that was statistically significant, either the two-sample $t$-test (for normally distributed samples) or Mann–Whitney $U$ test (for non-normally distributed samples) was used. Sample normality was assessed using the Lilliefors test. Means or medians of multiple subject groups were compared using either the one-way Analysis of Variance (ANOVA) (for normal samples) or Kruskal–Wallis test (for non-normal samples), and for comparisons with $p < 0.05$, a post hoc Tukey–Kramer multiple comparison was employed to determine significantly different pairs of groups. For data from the three longitudinal time points of Novice, repeated measures ANOVA (normal samples) or the non-parametric Friedman's test (non-normal sample) was also used to assess significance. For assessment of the correlation strength between two variables, the Pearson's correlation coefficient ($r$) was calculated. All statistical tests were executed using functions in the Statistics Toolbox of Matlab (R2019b). Statistical hypotheses were rejected at 5% significance.

**Reporting summary.** Further information on research design is available in the Nature Research Reporting Summary linked to this article.

## Data availability
The data that support the findings of this study are available at Open Science Framework at https://osf.io/pa6sn/?view_only=55395c49170d4021867e1eaa0f293d85. Source data are provided with this paper.

## Code availability
All code developed in the analysis are available to any researcher upon request sent to the corresponding authors.

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

## Acknowledgements

We thank Vittorio Caggiano for comments on versions of this manuscript. V.C.K.C., B.M.F.C., and S.C.W.H. were supported by The CUHK Faculty of Medicine Faculty Innovation Award FIA2016/A/04 and Group Research Scheme NL/JW/rc/grs1819/0426/19hc to V.C.K.C., and The Hong Kong Research Grants Council Grants 24115318 (ECS) and CUHK R4022-18 (RIF) to V.C.K.C. R.T.H.C., J.H.Z. and Z.Y.S.C. were supported by The Hong Kong Innovation and Technology Commission University-Industry Collaboration Programme Grant UIM/343 to R.T.H.C. C.Y.C. was supported by The Hong Kong Polytechnic University.

## Author contributions

V.C.K.C., B.M.F.C., and R.T.H.C. contributed to the study design; V.C.K.C., B.M.F.C., J.H.Z., Z.Y.S.C., S.C.W.H., and R.T.H.C. contributed to the collection of experimental data; V.C.K.C., B.M.F.C, J.H.Z., Z.Y.S.C, and R.T.H.C. contributed to the analysis of the data; V.C.K.C., B.M.F.C., J.H.Z., C.Y.C., and R.T.H.C. contributed to writing the manuscript.

## Competing interests

V.C.K.C. is a shareholder and otherwise uncompensated advisor of Pureform Technology Ltd. (Hong Kong). B.M.F.C. is a shareholder and otherwise uncompensated CEO of Pureform Technology Ltd. (Hong Kong). R.T.H.C. has received grant support (as Principal Investigator) from the University-Industry Collaboration Programme (project UIM/343) of the Innovation and Technology Commission, Government of the Hong Kong Special Administrative Region, China. J.H.Z., Z.Y.S.C., S.C.W.H. and C.Y.C. have no competing interest to declare.
