## [Peer Review File · Nature Communications]

Reviewers' Comments:

Reviewer #1:

Remarks to the Author:

This is a very well written manuscript tackling the issue of development and training on muscle synergies. The authors considered the effect of development and training on synergies by studying the synergies for running in preschoolers and adults from sedentary subjects and marathoners, with cross-sectional and longitudinal comparisons. The experimental work has been carefully performed and described. However there are problems with the analysis and interpretation of the results. The authors should address the following issues in order to convince the reader that what they report is correct and complete.

Major comments

1) An important point of this work is the cross-sectional comparison of the muscle synergies. Since muscle synergies generally depend on locomotor speed, a potential caveat is represented by the different speeds of the different groups. Children ran over-ground at self-selected speeds, but I couldn't find a specification of their speeds. Absolute speed values should be normalized to take into account the different size of participants, especially for children. The reported mean speed values for the adults were: sedentary: 6.2 ± 0.9 km/h; novice: 6.8 ± 0.9 km/h; experienced: 7.5 ± 1.3 km/h; elite: 12 km/h. The authors should assess the potential effect of the different speeds on muscle synergies.

2) Related to the above point, in the manuscript I couldn't find any comparison between the present results on untrained adults and the results obtained in previous work. There have been several publications on this topic (none of which I found cited here). In particular, Yokoyama et al. (2016; 2017) showed that human locomotor networks may have speed dependency, consistent with speed control mechanisms observed in insects and in vertebrates. Interestingly, they observed that the number of extracted locomotor modules (using NMF) increases with increasing speed. In 2016, Yokoyama et al. also compared different extracted sets of modules between non-runners and runners (>5 years of experience). They suggested that "the acquisition of novel locomotor movement following long-term training is achieved via the reorganization of locomotor networks consisting of existing locomotor modules". Since this is a key argument of the present manuscript, I think that Yokoyama's results should be compared and discussed in the present manuscript.

3) Another major concern is about the double integration of vertical ground reaction force (vGRF) used to assess the biomechanical performance. The authors apparently realized the vGRFs' integration only during the stance phase. The integration constant was thus selected to fix the position of the COM at the same position at the begin (landing) and end (take-off) of the stance phase (according to Fig. 1). Even if the Blickhan's spring-mass model (1989) assumes the same height and velocity of the center of mass at landing and take-off, these authors pointed out that 'animals in general do not have a similar takeoff and landing velocity as assumed in the model. They take off with straightened legs and land with bent leg, and the leg has to be stiffer during landing than during takeoff'. The deviation from the symmetrical model has been documented (see for example Cavagna 2006 or Maykranz and Seyfarth, 2014). Conversely, using kinematics data Dalleau et al. (1998) observed that the height of the COM was about the same at landing and take-off. This symmetry is most likely due to the fact that their subjects ran at a speed (18,3 km/h) much greater than in the present study (between 6 and 12 km/h): the speed tends to reduce the asymmetry between landing and take-off (Cavagna, 2006), as the role of the tendon becomes privileged.

In order to avoid the biased hysteresis, the integration constant must be set on the assumption that the average velocity of the center of mass is nil over at least one stride, as it has been done in the force-platform experiment reviewed by McMahon and Cheng (1990). This modification may change completely the authors' estimation of run efficiency and in turn the correlation with the

presence or absence of specific muscle-synergy merging patterns.

Minor comments:

- Inconsistent use of km/h or m/s (see for example fig S4 and Methods section)
- Change 'toddlers' (in Fig 5) in 'preschoolers' (as in the other figures)
- Why the running efficiency was not evaluated in preschoolers?

References

- T. A. McMahon, G. C. Cheng, The mechanics of running: how does stiffness couple with speed? *J. Biomech.* 23 Suppl 1, 65–78 (1990).
- G. Dalleau, A. Belli, M. Bourdin, J. R. Lacour, The spring-mass model and the energy cost of treadmill running. *Eur. J. Appl. Physiol. Occup. Physiol.* 77, 257–263 (1998).
- R. Blickhan, The spring-mass model for running and hopping. *J. Biomech.*, 22, 1217–1227 (1989).
- D. Maykranz, A. Seyfarth, Compliant ankle function results in landing-take off asymmetry in legged locomotion. *J. Theor. Biol.* 349, 44–49 (2014)
- G.A Cavagna, The landing-take-off asymmetry in human running. *J. Exp. Biol.* 209, 4051–4060 (2006)
- H. Yokoyama, T. Ogawa, N. Kawashima, M. Shinya, K. Nakazawa. Distinct sets of locomotor modules control the speed and modes of human locomotion. *Sci Rep* 6: 36275, (2016).
- H. Yokoyama, T. Ogawa, M. Shinya, N. Kawashima, K. Nakazawa Speed dependency in α -motoneuron activity and locomotor modules in human locomotion: indirect evidence for phylogenetically conserved spinal circuits. *Proc Biol Sci* 284: 20170290, (2017).
- D. J. Clark, L. H. Ting, F. E. Zajac, R. R. Neptune, S. A. Kautz, Merging of healthy motor modules predicts reduced locomotor performance and muscle coordination complexity post- stroke. *J. Neurophysiol.* 103, 844–857 (2010).

Reviewer #3:

Remarks to the Author:

NCOMMS-19-428952

Reshaping muscle synergies through fractionation and merging during development and training of human runners

by Vincent C. K. Cheung, Ben M. F. Cheung, Janet H. Zhang, Zoe Y. S. Chan, Sophia C. W. Ha, Chao-Ying Chen and Roy T. H. Cheung

In this paper, the authors addressed a critically important issue of motor control, i.e. plasticity of muscle synergies in human. The design and the results of this study are very original. The data are also original and analyzed deeply in a convincing way. I believe this study is valuable for neuroscientists, sports scientists and people in the wider field. I enjoyed reviewing this paper. I have only one comment to strengthen the authors' findings.

Specific comments:

Major

1) Page 7, lines 2-4: the authors stated that "Thus, merging of muscle-synergies during running training may be understood as a process that assigns multiple original W-encoding neuronal networks to be driven by one of the original oscillators (Fig. 7C)."

If that is the case, we would like to know how the innervation from the original C1 to W2 in Fig. 7C is newly established or reactivated for merging synergies during training. Furthermore, we would like to know the role of the reorganization mechanism for the reverse direction in development (i.e. fractionation of synergies). You may want to discuss these issues in Discussion.

Specific comments:

Minor

1) p 17, line 16: "when an R combination was present (D) or absent (E)" -> "when an R combination was absent (D) or present (E)"

Comments from Reviewer No. 1

This is a very well written manuscript tackling the issue of development and training on muscle synergies. The authors considered the effect of development and training on synergies by studying the synergies for running in preschoolers and adults from sedentary subjects and marathoners, with cross-sectional and longitudinal comparisons. The experimental work has been carefully performed and described. However there are problems with the analysis and interpretation of the results. The authors should address the following issues in order to convince the reader that what they report is correct and complete.

Major comments

- 1) An important point of this work is the cross-sectional comparison of the muscle synergies. Since muscle synergies generally depend on locomotor speed, a potential caveat is represented by the different speeds of the different groups. Children ran over-ground at self-selected speeds, but I couldn't find a specification of their speeds. Absolute speed values should be normalized to take into account the different size of participants, especially for children. The reported mean speed values for the adults were: sedentary: 6.2 ± 0.9 km/h; novice: 6.8 ± 0.9 km/h; experienced: 7.5 ± 1.3 km/h; elite: 12 km/h. The authors should assess the potential effect of the different speeds on muscle synergies.
- 2) Related to the above point, in the manuscript I couldn't find any comparison between the present results on untrained adults and the results obtained in previous work. There have been several publications on this topic (none of which I found cited here). In particular, Yokoyama et al. (2016; 2017) showed that human locomotor networks may have speed dependency, consistent with speed control mechanisms observed in insects and in vertebrates. Interestingly, they observed that the number of extracted locomotor modules (using NMF) increases with increasing speed. In 2016, Yokoyama et al. also compared different extracted sets of modules between non-runners and runners (>5 years of experience). They suggested that "the acquisition of novel locomotor movement following long-term training is achieved via the reorganization of locomotor networks consisting of existing locomotor modules". Since this is a key argument of the present manuscript, I think that Yokoyama's results should be compared and discussed in the present manuscript.
- 3) Another major concern is about the double integration of vertical ground reaction force (vGRF) used to assess the biomechanical performance. The authors apparently realized the vGRFs' integration only during the stance phase. The integration constant was thus selected to fix the position of the COM at the same position at the begin (landing) and end (take-off) of the stance phase (according to Fig. 1). Even if the Blickhan's spring-mass model (1989) assumes the same height and velocity of the center of mass at landing and take-off, these authors pointed out that 'animals in general do not have a similar takeoff and landing velocity as assumed in the model. They take off with straightened legs and land with bent leg, and the leg has to be stiffer during landing than during takeoff'. The deviation from the symmetrical model has been documented (see for example Cavagna 2006 or Maykranz and Seyfarth, 2014). Conversely, using kinematics data Dalleau et al. (1998) observed that the height of the COM was about the same at landing and take-off.

This symmetry is most likely due to the fact that their subjects ran at a speed (18,3 km/h) much greater than in the present study (between 6 and 12 km/h): the speed tends to reduce the asymmetry between landing and takeoff (Cavagna, 2006), as the role of the tendon becomes privileged. In order to avoid the biased hysteresis, the integration constant must be set on the assumption that the average velocity of the center of mass is nil over at least one stride, as it has been done in the force-platform experiment reviewed by McMahon and Cheng (1990). This modification may change completely the authors' estimation of run efficiency and in turn the correlation with the presence or absence of specific muscle-synergy merging patterns.

Minor comments:

- Inconsistent use of km/h or m/s (see for example fig S4 and Methods section)
- Change 'toddlers' (in Fig 5) in 'preschoolers' (as in the other figures)
- Why the running efficiency was not evaluated in preschoolers?

References

- T. A. McMahon, G. C. Cheng, The mechanics of running: how does stiffness couple with speed? *J. Biomech.* 23 Suppl 1, 65–78 (1990).
- G. Dalleau, A. Belli, M. Bourdin, J. R. Lacour, The spring-mass model and the energy cost of treadmill running. *Eur. J. Appl. Physiol. Occup. Physiol.* 77, 257–263 (1998).
- R. Blickhan, The spring-mass model for running and hopping. *J. Biomech.*, 22, 1217–1227 (1989).
- D. Maykranz, A. Seyfarth, Compliant ankle function results in landing-take off asymmetry in legged locomotion. *J. Theor. Biol.* 349, 44–49 (2014)
- G.A Cavagna, The landing-take-off asymmetry in human running. *J. Exp. Biol.* 209, 4051–4060 (2006)
- H. Yokoyama, T. Ogawa, N. Kawashima, M. Shinya, K. Nakazawa. Distinct sets of locomotor modules control the speed and modes of human locomotion. *Sci Rep* 6: 36275, (2016).
- H. Yokoyama, T. Ogawa, M. Shinya, N. Kawashima, K. Nakazawa Speed dependency in α motoneuron activity and locomotor modules in human locomotion: indirect evidence for phylogenetically conserved spinal circuits. *Proc Biol Sci* 284: 20170290, (2017).
- D. J. Clark, L. H. Ting, F. E. Zajac, R. R. Neptune, S. A. Kautz, Merging of healthy motor modules predicts reduced locomotor performance and muscle coordination complexity post-stroke. *J. Neurophysiol.* 103, 844–857 (2010).

Responses to Reviewer No. 1

This is a very well written manuscript tackling the issue of development and training on muscle synergies. The authors considered the effect of development and training on synergies by studying the synergies for running in preschoolers and adults from sedentary subjects and marathoners, with cross-sectional and longitudinal comparisons. The experimental work has been carefully performed and described. However there are problems with the analysis and interpretation of the results. The authors should address the following issues in order to convince the reader that what they report is correct and complete.

We thank the reviewer for the very constructive comments and suggestions. They have helped us improve our work very significantly. Below, we offer our responses to each of the points.

(1) An important point of this work is the cross-sectional comparison of the muscle synergies. Since muscle synergies generally depend on locomotor speed, a potential caveat is represented by the different speeds of the different groups. Children ran over-ground at self-selected speeds, but I couldn't find a specification of their speeds. Absolute speed values should be normalized to take into account the different size of participants, especially for children. The reported mean speed values for the adults were: sedentary: 6.2 ± 0.9 km/h; novice: 6.8 ± 0.9 km/h; experienced: 7.5 ± 1.3 km/h; elite: 12 km/h. The authors should assess the potential effect of the different speeds on muscle synergies.

We thank the reviewer for pointing out this important potential caveat on data interpretation. In this study, we reckon that to sample and study what the runners may have acquired and internalized into their motor system over development or training, it is more reasonable to collect data during their most natural forms of running. Thus, we decided to record our data at the runners' self-selected preferred speed. But both age and running experience correlate with the preferred speed. Indeed, in our data, the more experienced the adult subjects, the higher the preferred speed (Fig. S6A). Because of these correlations, it is true that some of the between-group muscle-synergy changes described here may result from variations of running speed. But here, we think that synergy fractionation and merging are more likely processes primarily related to development and training-induced changes in running efficiency. Our interpretation rests on the following observations:

- (1) When our three indicators of between-group muscle-synergy changes – the dimensionality, vector sparseness, and Merging Index – were regressed against the preferred running speed, statistically significant or nearly significant correlations were observed (unsurprisingly, because age/training correlate with speed, and these indices correlate with age/training). However, regardless of whether the regression was performed over just adults or preschoolers plus adults, the strength of correlation was low for all indices ($|r| = 0.20-0.36$) (Fig. S6B-D). When the correlations were performed within each subject group, none was significant ($p = 0.15-0.97$). Thus, either the running speed is not the sole determinant of synergy changes, or synergy changes are driven by other variables that loosely correlate with the preferred speed.
- (2) The presence or absence of the 5 biomechanically-relevant synergy merging combinations we identified separated the subjects with high and low running efficiency across groups with different preferred speeds. Indeed, for 3 of the 5 combinations, the preferred speeds of those possessing the combination were *not* statistically different from the speeds of those without them (Fig. S6E); for 1 of them, the difference was barely significant ($p = 0.043$) (Fig. S6E). Thus, these synergy-merging patterns are more related to efficiency than to running speed.
- (3) After normalizing the running speed to body height, the preschoolers' speed was similar to the elites' speed (Fig. S6A). If synergy fractionation from the preschoolers (Presch) to sedentary adults (Sedent) and then merging from Sedent to Elite is just a reflection of the decrease and then increase of the running speed, one would expect the

fractionation patterns to be identical to the merging combinations. But this is not what we found (Fig. S3A; Table S2 and Table S3).

- (4) Most importantly, changes of dimensionality, vector sparseness, and Merging Index were observed across the three longitudinal time points of the Novice subjects, who ran at the same preferred speed across the time points.

In the revised manuscript, we have described the above considerations in a new subsection of Discussion, ‘Some considerations on running speed’ (p. 9).

Concerning the running speeds of the preschoolers, in this revised manuscript we have specified both the absolute (in km/h) and body-height-normalized average self-selected running speed of the preschoolers (p. 24).

Concerning normalization of speed values, in the revised Materials and Methods, we have specified both the absolute (in km/h) and body-height (B. H.)-normalized running speeds of all groups (p. 24). All analyses in this version involving running speed were performed on the height-normalized speed (in B. H./s). These include our estimation of running efficiency values (Fig. S4A) and the correlations of our three indicators of muscle-synergy changes against the preferred running speed (Fig. S6).

(2) Related to the above point, in the manuscript I couldn’t find any comparison between the present results on untrained adults and the results obtained in previous work. There have been several publications on this topic (none of which I found cited here). In particular, Yokoyama et al. (2016; 2017) showed that human locomotor networks may have speed dependency, consistent with speed control mechanisms observed in insects and in vertebrates. Interestingly, they observed that the number of extracted locomotor modules (using NMF) increases with increasing speed. In 2016, Yokoyama et al. also compared different extracted sets of modules between non-runners and runners (>5 years of experience). They suggested that “the acquisition of novel locomotor movement following long-term training is achieved via the reorganization of locomotor networks consisting of existing locomotor modules”. Since this is a key argument of the present manuscript, I think that Yokoyama’s results should be compared and discussed in the present manuscript.

We thank the reviewer once again for directing us to the studies of Yokoyama *et al.* (2016, 2017) on the relationship between running speed and muscle synergies. We have studied the two relevant papers by Yokoyama *et al.* in detail. The primary conceptual goal of their studies was to answer how the muscle synergies for running change with the running speed, and thus, the synergies of the non-runners and runners were not explicitly compared. But we fully agree with the reviewer that it should be instructive to discuss and compare their results with ours. In the supplementary text (section 5, titled ‘Comparison with data reported in Yokoyama *et al.* (2016)’), we have provided a detailed comparison of our data with theirs. A close examination of the synergies of the non-runners and runners in Yokoyama *et al.* reveals that some of our efficiency-enhancing merging combinations may actually be present in their runners but not non-runners, and that our efficiency-reducing combinations may be present in their non-runners but not runners (see supplementary text, section 5, for details). These comparisons, though qualitative in nature, nonetheless argue for the consistency of our results with those of Yokoyama *et al.*, thus further supporting the validity of our finding.

In the revised manuscript, we have not only cited both papers by Yokoyama *et al.*, but also multiple previous studies on the relationship between motor training and muscle synergies. As the reviewer pointed out, the concept that the CNS exploits and modifies existing motor patterns (rather than assembling new patterns *de novo*) for achieving motor skill learning has been implied or suggested in multiple previous works, including those of Yokoyama *et al.* Sawers, Allen and Ting (2015), for example, have shown that professional ballet dancers modify their muscle synergies for walking so that the same set of synergies are used for both daily walking and challenging postural tasks. Kargo and Nitz (2003) have demonstrated in rats that during early skill learning of a forelimb task, the muscle weightings of specific synergies are gradually updated, and this fine-tuning also correlates with changes in the firing rates of motor cortical neurons. Here, by comparing the synergies from runners with a wide range of training experience (0-30 years), we argue further that muscle-synergy merging can be a general mechanism that the motor system employs – at least for running – to generate new patterns from pre-existing synergies, and situate this mechanism as one that is opposite to the process that drives developmental changes of synergies (i.e., fractionation). We also explicitly relate specific merging combinations to an increased or decreased energetic efficiency of running, and show how these combinations may be related to training. Additionally, we also suggest how training can fail if merging happens to the “wrong” efficiency-reducing combination of muscle synergies (e.g., *S0-4+5+7* in the Exp group). We believe we have extended the previous results in original and meaningful ways.

In the revised manuscript, we have emphasized the contributions of the previous works and the novelty of our results in the last subsection of Discussion, ‘Outcomes of motor development as building blocks of subsequent training’ (p. 10).

Concerning the relationship between running speed and the number of synergies, as shown in our Fig. S6B, across our subject groups the number of synergies tended to decrease as the preferred running speed increased. But as the reviewer pointed out, in Yokoyama *et al.* (2016), in both male runners and non-runners, the number of synergies increased as running speed increased. We think our finding is not necessarily inconsistent with this previous result. It is plausible that when running at the self-selected, preferred speed, the motor system employs a set of designated, “default mode” synergies whose dimensionality decreases with higher preferred speeds and training experience; but when running beyond the preferred speed, additional muscle synergies are employed. In fact, Yokoyama *et al.* (2016) arrived at their conclusion by testing the subjects over their full speed ranges and by comparing the synergies derived from their minimum, moderate, to maximum speeds, respectively. On the other hand, we tested ours only at their preferred speeds, also likely to be at the lower end of their natural speed ranges. In other words, the preferred speed and instantaneous running speed may be two separate determinants of the number of synergies employed. We have provided the above consideration in the revised Discussion, under the subsection ‘Some considerations on running speed’ (p. 9).

(3) Another major concern is about the double integration of vertical ground reaction force (vGRF) used to assess the biomechanical performance. The authors apparently realized the vGRFs’

integration only during the stance phase. The integration constant was thus selected to fix the position of the COM at the same position at the begin (landing) and end (take-off) of the stance phase (according to Fig. 1). Even if the Blickhan's spring-mass model (1989) assumes the same height and velocity of the center of mass at landing and take-off, these authors pointed out that 'animals in general do not have a similar takeoff and landing velocity as assumed in the model. They take off with straightened legs and land with bent leg, and the leg has to be stiffer during landing than during takeoff'. The deviation from the symmetrical model has been documented (see for example Cavagna 2006 or Maykranz and Seyfarth, 2014). Conversely, using kinematics data Dalleau et al. (1998) observed that the height of the COM was about the same at landing and take-off. This symmetry is most likely due to the fact that their subjects ran at a speed (18,3 km/h) much greater than in the present study (between 6 and 12 km/h): the speed tends to reduce the asymmetry between landing and takeoff (Cavagna, 2006), as the role of the tendon becomes privileged. In order to avoid the biased hysteresis, the integration constant must be set on the assumption that the average velocity of the center of mass is nil over at least one stride, as it has been done in the force-platform experiment reviewed by McMahon and Cheng (1990). This modification may change completely the authors' estimation of run efficiency and in turn the correlation with the presence or absence of specific muscle-synergy merging patterns.

We fully agree with the reviewer that it should be much more accurate to estimate CoM displacement by determining the integration constant through the assumption that the average CoM velocity is nil over each stride (see the revised Fig. 1C). We recalculated all energy loss values using this method and re-estimated the running efficiency based on the new values. When these new estimates were correlated with the synergy merging patterns, we reproduced 4 of the 6 original patterns derived from the old values (*S0-7+11*, *5+6+8*, *5+6+12*, and *4+5+7*), and found 1 new efficiency-reducing pattern (*S0-3+12*). Thus, in this updated version a total of 5 energetically-relevant synergy merging combinations are described.

In the revised Materials and Methods, we have updated the section, 'Biomechanical analysis' (p. 25) to reflect our use of this more correct procedure.

Inconsistent use of km/h or m/s (see for example fig S4 and Methods section)

All absolute speed values are now cited in km/h. Speeds in Fig. S4 and Fig. S6 are plotted in B. H./s.

Change 'toddlers' (in Fig 5) in 'preschoolers' (as in the other figures)

Fig. 5 was remade, and the original 'toddler' label was replaced with 'preschooler'.

Why the running efficiency was not evaluated in preschoolers?

Even though our preschoolers ran over-ground on a runway with force plates embedded midway of the path, it was very difficult to ensure that they stepped on the plates correctly during the recording session. Thus, we did not have enough high-quality force data from the preschoolers for reliable estimations of both the ground reaction force and vertical

displacement. Ideally it would be easier to collect force data during treadmill running, but we did not conduct any treadmill session for preschoolers to ensure their safety. In our revised Materials and Methods we included a justification of why we did not use the preschooler force data (p. 24).

Comments from Reviewer No. 3

NCOMMS-19-428952

Reshaping muscle synergies through fractionation and merging during development and training of human runners

by Vincent C. K. Cheung, Ben M. F. Cheung, Janet H. Zhang, Zoe Y. S. Chan, Sophia C. W. Ha, Chao-Ying Chen and Roy T. H. Cheung

In this paper, the authors addressed a critically important issue of motor control, i.e. plasticity of muscle synergies in human. The design and the results of this study are very original. The data are also original and analyzed deeply in a convincing way. I believe this study is valuable for neuroscientists, sports scientists and people in the wider field. I enjoyed reviewing this paper. I have only one comment to strengthen the authors' findings.

Specific comments:

Major

1) Page 7, lines 2-4: the authors stated that "Thus, merging of muscle-synergies during running training may be understood as a process that assigns multiple original W-encoding neuronal networks to be driven by one of the original oscillators (Fig. 7C)." If that is the case, we would like to know how the innervation from the original C1 to W2 in Fig. 7C is newly established or reactivated for merging synergies during training. Furthermore, we would like to know the role of the reorganization mechanism for the reverse direction in development (i.e. fractionation of synergies). You may want to discuss these issues in Discussion.

Minor

1) p 17, line 16: "when an R combination was present (D) or absent (E)" -> "when an R combination was absent (D) or present (E)"

Shinji Kakei, MD, PhD.

Movement disorders project

Tokyo Metropolitan Institute of Medical Science

Tokyo Japan

Responses to Reviewer No. 3

In this paper, the authors addressed a critically important issue of motor control, i.e. plasticity of muscle synergies in human. The design and the results of this study are very original. The data are also original and analyzed deeply in a convincing way. I believe this study is valuable for neuroscientists, sports scientists and people in the wider field. I enjoyed reviewing this paper. I have only one comment to strengthen the authors' findings.

We thank Dr. Kakei for his appreciation of our work! We have been encouraged to work even harder to improve and strengthen this manuscript. Below, we offer our responses to each of your points.

Page 7, lines 2-4: the authors stated that “Thus, merging of muscle-synergies during running training may be understood as a process that assigns multiple original W-encoding neuronal networks to be driven by one of the original oscillators (Fig. 7C).” If that is the case, we would like to know how the innervation from the original C1 to W2 in Fig. 7C is newly established or reactivated for merging synergies during training. Furthermore, we would like to know the role of the reorganization mechanism for the reverse direction in development (i.e. fractionation of synergies). You may want to discuss these issues in Discussion.

We fully agree that a discussion on how synergy merging and fractionation may be achieved through a reorganization mechanism should strengthen the manuscript. As we formulated a plausible mechanism, we were inspired by the recent results of Kawai and Ölveczky *et al.* (2015, *Neuron*) that, at least for rodents, the motor cortical areas are required for the learning of non-dexterous skills, but not the skills’ execution once they have been learned. In this regard, the motor cortex may function as a “tutor” that directs subcortical plasticity for assembling a proper motor sequence critical for executing the learned skill. We speculate that muscle synergy merging and fractionation may be similarly accomplished through a cortical tutor. During merging, the new connection between the $C_1(t)$ oscillator and the W_2 synergy-encoding network in our example (Fig. 7C) may be reinforced by descending cortical inputs that synchronously activate both C_1 and W_2 (Fig. 8A). During early training, such synchronous activation may initially occur by chance as the runner explores the motor-command space through motor variability; subsequently, it can be reinforced by reward signals or other afferents that signify the metabolic cost or sense of effort. This mechanism, speculative as it is, is nonetheless consistent with the earlier finding of Kargo and Nitz (2003, *J. Neurosci.*) that, in rodents, during early skill learning changes of motor cortical firing correlate with the changes of specific muscle synergies. As to fractionation, it can likewise be achieved through a cortical tutor that synchronizes a network being driven by an “idle” oscillator to the motoneurons of a subset of muscles within an existing synergy (Fig. 8B).

Our model above implies that each of the multiple oscillating networks that generate rhythmic activity bursts at different phases of the locomotor cycle can be flexibly reassigned to drive different subsets of the downstream synergy-encoding networks. This flexibility of the relationship between the encoded temporal and spatial patterns may well be a general mechanism responsible for the plasticity of locomotor patterns.

In this revised manuscript, we have included a new section in Discussion (‘Plausible neural mechanisms of synergy merging and fractionation’, p. 9) that describes our hypothetical model above. We have also included a new figure (Fig. 8) that illustrates our model. We have decided not to include these new schematics in the original Fig. 7 because the model of cortical tutoring is speculative in nature while the schematic in Fig. 7C is grounded on our analysis of the synergies’ temporal coefficients.

P 17, line 16: “when an R combination was present (D) or absent (E)” -> “when an R combination was absent (D) or present (E)”

In this revised manuscript, we have redesigned Fig. 6 based on new estimates of running efficiency determined by a more accurate method of estimating energy loss (see responses to Reviewer No. 1 above). To avoid inadvertent mistakes and typos, in our revision, we were extra mindful of the use of “absence” and “presence”.

Reviewers' Comments:

Reviewer #1:

Remarks to the Author:

The ms has been thoroughly revised following the reviewers' recommendations. The answer to the question of the potential effect of speed on muscle synergies is convincing. The authors' hypothesis that "It is plausible that when running at the self-selected preferred speed, one employs a set of designated, default mode synergies" is interesting, although it deserves further scrutiny.

The authors did not pursue much further biomechanical analysis, as we asked in the first round. Nevertheless, their approach to the problem is satisfactory. They used a complex metric to highlight a simple landing-take off asymmetry, the hysteresis (only depending on potential energy) corresponds per se to the modification of the COM position at foot-contact and toe-off. This is a good way to illustrate the deviation from an elastic rebound.

Reviewer #3:

Remarks to the Author:

The authors have answered for all my comments perfectly.

Responses to Reviewers

Reviewer No. 1

The ms has been thoroughly revised following the reviewers' recommendations. The answer to the question of the potential effect of speed on muscle synergies is convincing. The authors' hypothesis that "It is plausible that when running at the self-selected preferred speed, one employs a set of designated, default mode synergies" is interesting, although it deserves further scrutiny.

We would like to once again express our gratitude to Prof. Lacquaniti and Dr. Dewolf for your review of our manuscript. Your constructive comments have really helped us improve the quality of our work significantly.

We are glad that you have found our additional analyses on assessing the potential effect of speed on synergies to be convincing. We agree that whether the dimensionality of the EMGs depends on both the preferred speed and instantaneous speed for both walking and running warrants further investigation. In Discussion, we have stressed that further experiments will be needed to clarify these potential dependencies.

The authors did not pursue much further biomechanical analysis, as we asked in the first round. Nevertheless, their approach to the problem is satisfactory. They used a complex metric to highlight a simple landing-take off asymmetry, the hysteresis (only depending on potential energy) corresponds per se to the modification of the COM position at foot-contact and toe-off. This is a good way to illustrate the deviation from an elastic rebound.

Francesco Lacquaniti and Arthur Dewolf

We are likewise glad that you have found our biomechanical analysis to be satisfactory. We agree totally that it would be very interesting and important to follow up this work with additional biomechanical analyses especially for the preschooler group. Analyses along the lines summarized in your very recent review paper (*Front. Bioengin. Biotech.* **8**, art. 473, May 2020) should be very fruitful. In the revised text, we have cited the above paper as a reference.

Reviewer No. 3

The authors have answered for all my comments perfectly.

We would once again like to express our gratitude to Prof. Kakei for reviewing our revised manuscript. Your constructive comments have helped us improve the quality of our work very significantly.